# Petal size is controlled by the MYB73/TPL/HDA19-miR159-CKX6 module regulating cytokinin catabolism in *Rosa hybrida*

Weikun Jing [1,2,3,5], Feifei Gong[1,5], Guoqin Liu[1,4,5], Yinglong Deng[1], Jiaqi Liu [1], Wenjing Yang[1], Xiaoming Sun[1], Yonghong Li[3], Junping Gao [1], Xiaofeng Zhou [1] ✉ & Nan Ma [1] ✉

The size of plant lateral organs is determined by well-coordinated cell proliferation and cell expansion. Here, we report that miR159, an evolutionarily conserved microRNA, plays an essential role in regulating cell division in rose (*Rosa hybrida*) petals by modulating cytokinin catabolism. We uncover that *Cytokinin Oxidase/Dehydrogenase6* (*CKX6*) is a target of miR159 in petals. Knocking down miR159 levels results in the accumulation of *CKX6* transcripts and earlier cytokinin clearance, leading to a shortened cell division period and smaller petals. Conversely, knocking down *CKX6* causes cytokinin accumulation and a prolonged developmental cell division period, mimicking the effects of exogenous cytokinin application. MYB73, a R2R3-type MYB transcription repressor, recruits a co-repressor (TOPLESS) and a histone deacetylase (HDA19) to form a suppression complex, which regulates *MIR159* expression by modulating histone H3 lysine 9 acetylation levels at the *MIR159* promoter. Our work sheds light on mechanisms for ensuring the correct timing of the exit from the cell division phase and thus organ size regulation by controlling cytokinin catabolism.

How the size of living organisms is regulated makes for a fascinating and attractive research question[1–3]. The final size of a given organ is carefully orchestrated by the complex integration of multiple intrinsic growth signals and external environmental cues. The growth of petals, just like other organs, goes through four phases from primordium to their final size: (1) founder cells are recruited to the primordium, (2) cell proliferation, (3) transition from cell proliferation to cell expansion, and (4) cell expansion. Final organ size is determined by a strict spatial and temporal coordination of these four events[1,4]. Organ size also reflects the integration of total cell number, which is determined by the number of cell divisions[5–8] and the timing of proliferation arrest[9–11], with cell size, which is defined by the duration of the cell expansion phase and the cell expansion rate[1,12,13].

Several phytohormones, such as auxin, gibberellin (GA), cytokinins, and brassinosteroids (BRs), coordinately control the dynamics of the so-called "cell cycle arrest front", which marks the transition from cell division to differentiation[14–18]. In Arabidopsis (*Arabidopsis thaliana*) leaves, most cells exit the division phase and begin expanding when the leaf blade reaches a certain size (about 10% of final size)[19,20]. ARGOS, encoded by an auxin-inducible gene, promotes cell proliferation by upregulating *AINTEGUMENTA* (*ANT*) and *ANT-LIKE6* (*AIL6*) expression, leading to the transcriptional activation of the D-type cyclin gene *CYCD3;1*[11,21,22]. AUXIN RESPONSE FACTOR8 (ARF8) interacts

[1]Beijing Key Laboratory of Development and Quality Control of Ornamental Crops, Department of Ornamental Horticulture, College of Horticulture, China Agricultural University, Beijing 100193, China. [2]Flower Research Institute of Yunnan Academy of Agricultural Sciences, Kunming, Yunnan 650205, China. [3]School of Food and Medicine, Shenzhen Polytechnic, Shenzhen, Guangdong 518055, China. [4]College of Agronomy, Henan Agricultural University, Zhengzhou 450002, China. [5]These authors contributed equally: Weikun Jing, Feifei Gong, Guoqin Liu. ✉e-mail: zhouxiaofeng@cau.edu.cn; ma_nan@cau.edu.cn

with the transcription factor BIG PETAL (BPEp) and synergistically restricts mitotic growth in petals by influencing proliferation and expansion[23,24]. Another auxin-responsive transcription factor, ARF2, suppresses the expression of *ANT* and *CYCD3;1* and thus inhibits proliferation. Notably, ARF2 activity is itself repressed by protein phosphorylation via the brassinosteroid-responsive kinase BR-INSENSITIVE2 (BIN2)[25,26], indicating that cross-talk between auxin and BRs contributes to cell proliferation and expansion. In parallel, GA induces the expression of *STUNTED* (*STU*), encoding a receptor-like cytoplasmic kinase that represses the expression of the cyclin-dependent kinase inhibitor genes *SIAMESE* (*SIM*) and *SIAMESE-RELATED1* (*SMR1*), which encode pro-expansion factors by stimulating endoreduplication[27]. In maize (*Zea mays*) leaves, GA maintains cell proliferation, and a peak in GA concentration marks the boundary between cell proliferation and expansion zones[15,28,29].

Cytokinins play vital roles in plant development by controlling cell division. Cytokinin metabolism and signal transduction are well understood[30,31]. Zeatin and isopentenyladenine-type cytokinins are considered to be the predominant forms of active cytokinins[32,33]. Steady-state levels of active cytokinins in a given organ are determined by the rates of biosynthesis and release of cytokinin nucleobase from their inactive conjugates and the rates of cytokinin degradation and inactivating conjugation[30]. Cytokinin oxidase/dehydrogenase (CKX) catalyzes the irreversible degradation of cytokinins[34,35]. In Arabidopsis, overexpressing *CKX1* to *CKX6* resulted in enhanced cytokinin breakdown and diminished activity of the vegetative and floral shoot apical meristems and leaf primordia[36]. Compared to the wild-type and single mutants, the *ckx3 ckx5* double mutant produced larger flowers due to a delayed onset of cellular differentiation and an extended developmental cell division period within the inflorescence and floral meristems[37].

microRNAs (miRNAs) are small RNAs of 20–24 nucleotides in length that play an essential role in developmental decisions[38]. Several miRNAs have been reported to control floral organ initiation, growth, and maturation. miR172 controls floral organ identity by binding to *APETALA2* (*AP2*) mRNA and triggering its degradation or blocking APETALA2 protein translation in Arabidopsis[39]. miR319a governs petal and stamen development by targeting *TCP4* mRNA encoding a member of the Teosinte branched1/Cincinnata/proliferating cell factor (TCP) family of transcription factors[40]. miR159, an evolutionarily conserved miRNA, is indispensable for normal floral organ growth[41]. The canonical miR159 target genes encode members of the GA-induced MYB (GAMYB) or GAMYB-like family, which are R2R3 MYB transcription factors present in most land plants and highly conserved[42–46]. Besides these conserved *GAMYB* targets, several other transcripts have been predicted to be potential miR159 targets in Arabidopsis, such as *1-AMINO-CYCLOPROPANE-1-CARBOXYLATE SYNTHASE8* (*ACS8*, At4g37770) and *COPPER/ZINC SUPEROXIDE DISMUTASE3* (*CSD3*, At5g18100)[47–50]. In tomato (*Solanum lycopersicum*), miR159 regulates leaf and flower development by targeting SGN-U567133, a non-canonical target encoding a nucleus-localized protein with a NOZZLE-like domain[51].

Our previous work predicted a *CKX* gene as a potential miR159 target in rose (*Rosa hybrida*)[52]. Here, we used rose petals as a model of lateral organ growth and discovered that miR159-mediated cytokinin clearance is required for controlling the duration of the cell division phase during the early petal growth period. Furthermore, we established that the MYB73–TOPLESS (TPL)–Histone Deacetylase19 (HDA19) repression complex developmentally determines *MIR159* expression by modulating histone H3 lysine 9 acetylation (H3K9ac) levels at the *MIR159* promoter. Importantly, mature miR159 targeted transcripts from the cytokinin oxidase/dehydrogenase gene *RhCKX6* in petals for mRNA cleavage. Knockdown of miR159 abundance resulted in rapid accumulation of *RhCKX6* transcripts, leading to cytokinin catabolism and a consequent shortened cell division period and

smaller petals. Together, our results reveal a previously unknown level of regulatory complexity for ensuring the correct timing of cell division and organ size via controlling cytokinin catabolism.

## Results

### miR159 controls the duration of the cell division phase and flower development by targeting *RhCKX6* transcripts in rose petals

Organ growth requires coordination between cell division and cell expansion[53,54]. During rose petal development, the adaxial and abaxial epidermal cells display distinct growth patterns. Adaxial cells continue to divide until the flower opens, making cell division a major contributor to adaxial petal area. Conversely, abaxial cells mostly stop division before bud opening, with cellular expansion becoming the primary contributor to abaxial petal area[55–57]. To understand the underlying control mechanism of cell division and cell expansion, we previously conducted a transcriptome deep sequencing (RNA-seq) analysis of short RNAs and transcripts from petals to investigate gene expression profiles during the cell division and expansion phases[52,58]. Based on our previous results, we defined stage 0 as "floral bud revealed" and stage 2 as "sepal completely unfolded". Microscopy observations indicated that abaxial epidermal cell division almost stops at stage 2, whereas cell expansion drove petal growth from stage 2 onward, as demonstrated by the rapid increase of petal size at this stage (Supplementary Fig. 1).

The above dataset revealed that miR159 abundance exhibits a sharp drop in petals from cell division (stage 0) to cell expansion (5 days after stage 0)[56,59,60]. Small RNA blot and reverse transcription quantitative PCR (RT-qPCR) analyses confirmed the pattern of miR159 abundance in petals, with an huge drop from d 1 to d 5 after stage 0 (Fig. 1a; Supplementary Fig. 2a). The precursor sequence of this rose *MIR159* (rhy-*MIR159*) shared high similarity with apple (*Malus domestica*) *MIR159* (mdm-*MIR159d/e/f*), especially over the stem region (Supplementary Fig. 2b–d).

To characterize the function of miR159 in rose petals, we knocked down miR159 levels using a *Tobacco rattle virus* (TRV)-mediated short tandem target mimic (STTM) approach[61,62] (Supplementary Fig. 3a). Notably, knocking down miR159 levels significantly shortened the petal cell division period (between stages 0 and 2), with $3.40 \pm 0.80$ days in TRV-*STTM159* and $5.10 \pm 0.70$ days in the TRV control, while having no effect on the cell expansion period between stages 2 and 5 (Fig. 1b–d). The final petal size (at fully opened stage 5) of TRV-*STTM159* lines was smaller than that of TRV controls (Fig. 1e). We recorded the number of adaxial and abaxial epidermal cells in TRV controls and TRV-*STTM159* lines: knocking down miR159 levels resulted in a slower increase in the number of adaxial and abaxial epidermal cells, particularly in abaxial epidermal cells, than in control petals (Fig. 1f, g). These results indicate that the smaller petal size of TRV-*STTM159* likely arose from fewer cells. Therefore, we hypothesized that miR159 affects the duration of cell proliferation in petals.

To explore the possible regulatory network involving miR159 during petal development, we compared the transcriptomes of TRV-*STTM159* and TRV petals. We identified 1358 differentially expressed genes (DEGs) (using the criteria $|Log_2[fold\ change]| \geq 1$ and $q$-value $\leq 0.05$ for TRV-*STTM159*/TRV), comprising 497 upregulated and 861 downregulated genes (Supplementary Fig. 3b; Supplementary Data 1). Kyoto Encyclopedia of Genes and Genomes (KEGG) pathway enrichment analysis of these DEGs indicated that terms associated with 'plant hormone signal transduction' and 'zeatin biosynthesis' are significantly enriched (Fig. 2a), suggesting that miR159 regulates phytohormone signaling, probably cytokinins. Notably, target prediction analysis identified 96 possible miR159 targets in the rose genome, including a CKX gene (RchiOBHmChr1g0319331) (https://www.zhaolab.org/psRNATarget/) (Supplementary Data 2 and Supplementary Fig. 4). Of all predicted targets, only

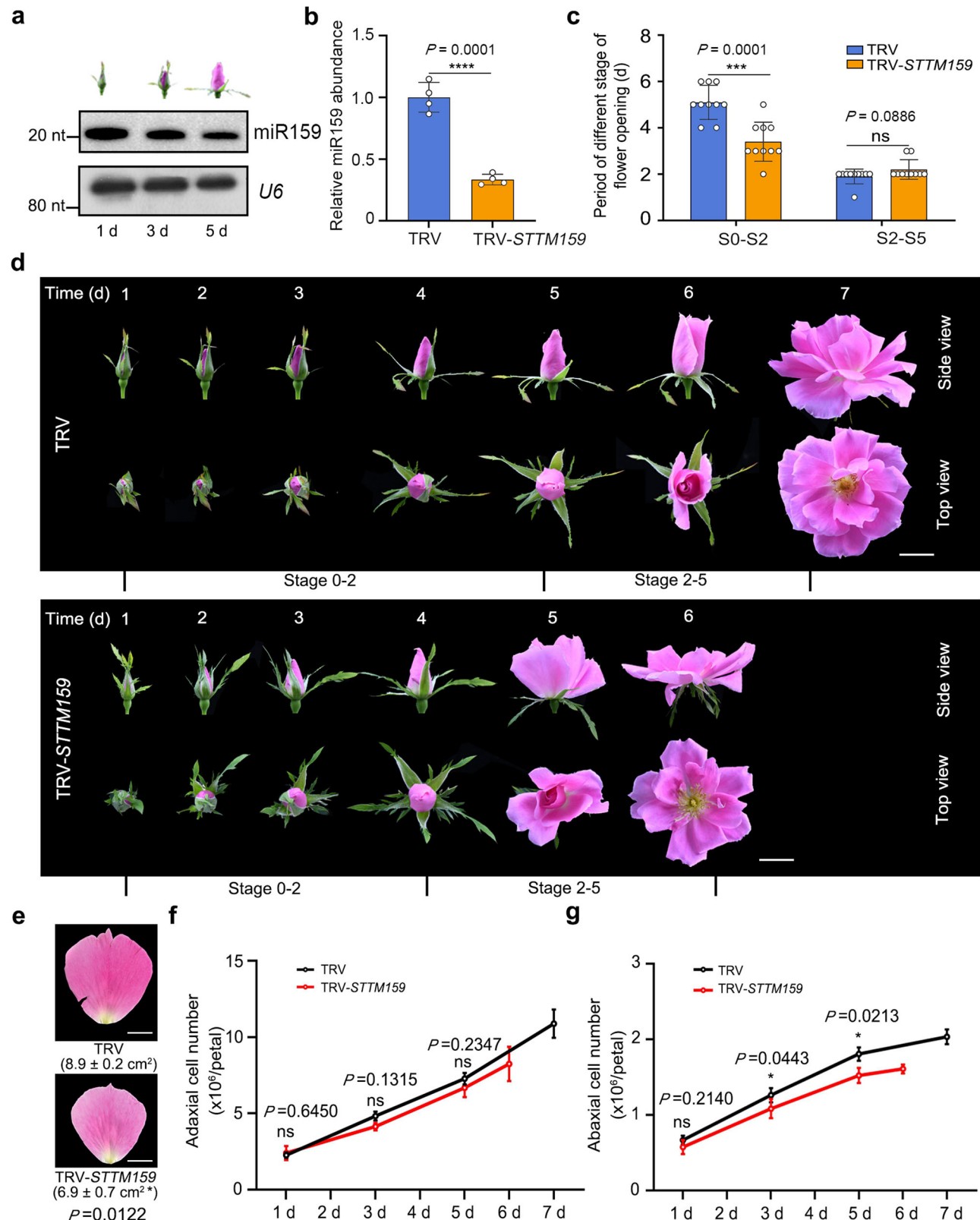

RchiOBHmChr1g0319331 showed a robust increase in transcript levels in TRV-*STTM159* lines compared to the TRV control, as would be expected for a true miR159 target (Fig. 2b–d). Moreover, RchiOBHmChr1g0319331 expression significantly increased from the cell division phase to the cell expansion phase, which was opposite to the pattern of miR159 abundance (Fig. 2e, f). Phylogenetic analysis showed that RchiOBHmChr1g0319331 encodes a CKX that is closely

related to CKX6 from Arabidopsis; we thus named the rose gene *RhCKX6* (Supplementary Fig. 5).

According to the target predictions above, the putative miR159 target site was located in the third exon of *RhCKX6*, 883–902 bp downstream of the translation start site of the open reading frame. We performed a 5′ RNA ligase-mediated rapid amplification of cDNA ends (RLM-RACE) assay to test the miR159-dependent cleavage of *RhCKX6*

**Fig. 1 | miR159 controls the duration of the cell division phase and flower opening in rose. a** Small RNA blot analysis of miR159 abundance in rose petals 1, 3, and 5 days after floral buds reached stage 0. Locked Nucleic Acids (LNA)-modified oligonucleotide probe was used to ensure the specificity of miR159. *U6* was used as an internal control. The experiment was performed three times. **b** Reverse transcription quantitative PCR (RT-qPCR) analysis of miR159 levels in sepals (3 days after stage 0) of TRV and TRV-*STTM159* plants. Data are shown as means ± SD (*n* = 4). 5 S rRNA was used as an internal control. Asterisks indicate statistically significant differences (two-sided Student's *t*-test; ****P* < 0.0001). **c, d** Flower opening progression of TRV and TRV-*STTM159* plants. The experiments were performed independently twice with similar results, and one representative set of results is shown. Scale bars, 2 cm. Data are shown as means ± SD (*n* = 10). Asterisks indicate statistically significant differences (two-sided Student's *t*-test; ****P* < 0.001). **e** Petal size of TRV control and TRV-*STTM159* plants at the fully opened stage. Data are shown as means ± SD (*n* = 3). The numbers below the images indicate petal size. Scale bar, 1 cm. Asterisks indicate statistically significant differences (two-sided Student's *t*-test; **P* < 0.05). **f, g** Cell number on the adaxial epidermis (**f**) and abaxial epidermis (**g**) in TRV control and TRV-*STTM159* petals from 1 day after floral stage 0 to the fully opened stage. Data are shown as means ± SD (*n* = 3). Asterisks indicate statistically significant differences (two-sided Student's *t*-test; **P* < 0.05).

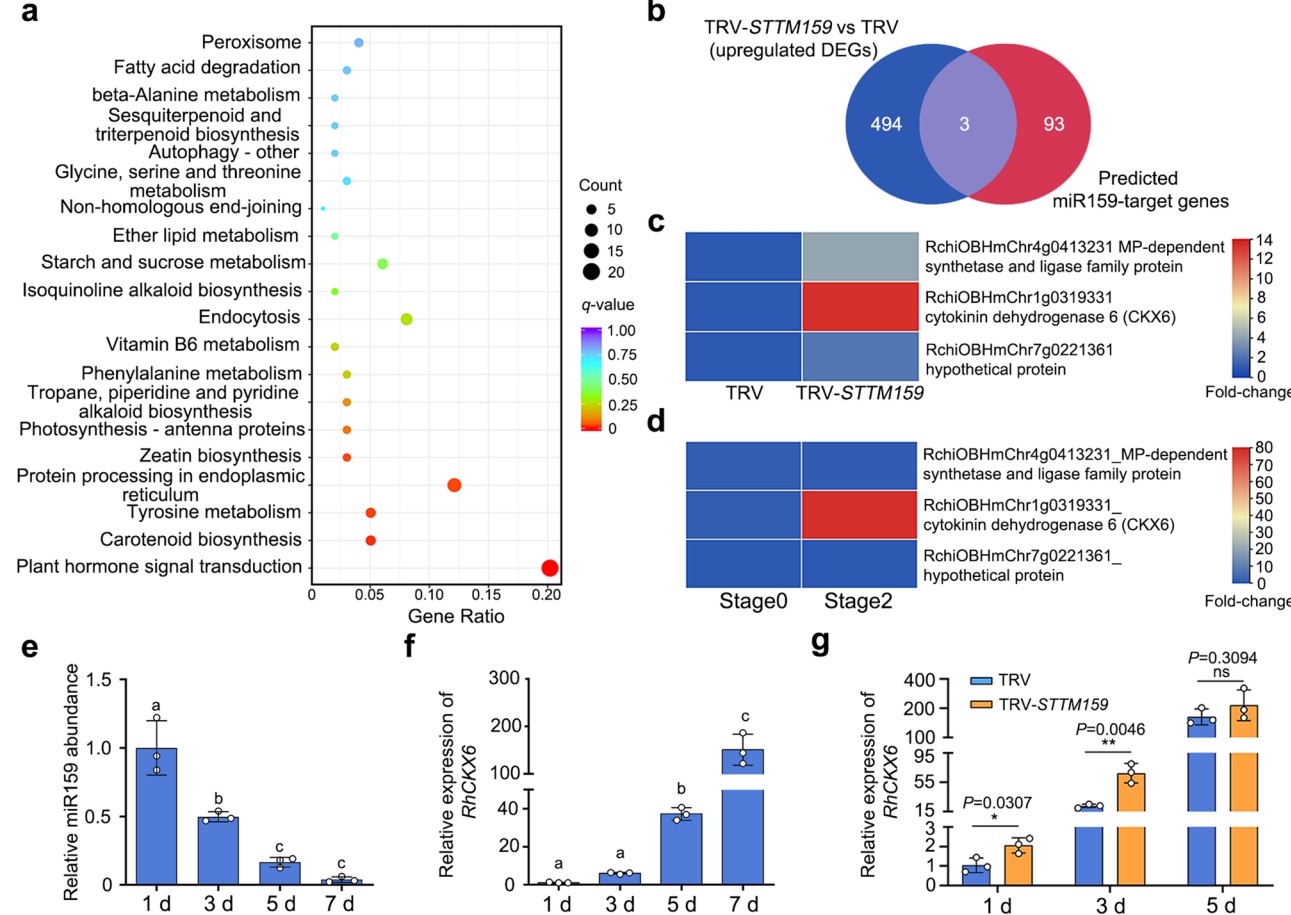

**Fig. 2 | miR159 influences transcript accumulation of cytokinin catabolism genes. a** Significantly enriched KEGG pathways in TRV-*STTM159* lines. Arrows indicated the hormone-related pathways. **b** Venn diagram showing the extent of overlap between differentially expressed genes (DEGs) in TRV-*STTM159* lines compared to TRV (absolute Log2[fold change] ≥ 1 and *q*-value ≤ 0.05) and predicted miR159-target genes. **c, d** Heatmap representation of transcript levels for possible miR159-targeted genes in TRV-*STTM159* and TRV plants (**c**) and stage 0 and stage 2 (**d**). **e, f** RT-qPCR analysis of miR159 abundance (**e**) and *RhCKX6* transcript levels (**f**) in rose petals at 1, 3, 5, and 7 days after stage 0. 5 S rRNA (**e**) or *RhUBI2* (**f**) was used as an internal control. Data are shown as means ± SD (*n* = 3). Different lowercase letters indicate significant differences according to one-way ANOVA with Tukey's multiple comparisons test (*P* < 0.05). **g** RT-qPCR analysis of *RhCKX6* transcript levels in petals of TRV control and TRV-*STTM159* at 1, 3, and 5 days after stage 0. *RhUBI2* was used as an internal control. Data are shown as means ± SD (*n* = 3). Asterisks indicate statistically significant differences (two-sided Student's *t*-test; **P* < 0.05; ***P* < 0.01; ns no significant difference).

transcripts in rose petals. As shown in Fig. 3a, we mainly detected cleavage at several sites within the region complementary to miR159. We constructed *RhCKX6*-Sensor and *RhCKX6m*-Sensor constructs by cloning the coding sequence of the green fluorescent protein (*GFP*) gene in-frame with an intact miR159-target region (for *RhCKX6*-Sensor) or a mutated miR159-target region (for *RhCKX6m*-Sensor) (Fig. 3a, b). We then co-infiltrated each sensor construct in *Nicotiana benthamiana* leaves with a construct overexpressing the miR159 precursor (*pre-MIR159*) and a viral silencing suppressor p19; importantly, we detected lower GFP fluorescence for the *pre-MIR159* + *RhCKX6*-Sensor combination compared to the sensor alone. By contrast, co-infiltrating *pre-MIR159* with *RhCKX6m*-Sensor did not influence fluorescence intensity

derived from *RhCKX6m*-Sensor (Fig. 3b, c). Immunoblot analysis confirmed the dependence of GFP abundance on miR159 (Fig. 3d). These results indicate that *RhCKX6* is an authentic target of miR159 in rose. Furthermore, we detected a predicted miR159-target site in the putative homolog *FvCKX6* from wild strawberry (*Fragaria vesca*). We observed miR159-mediated cleavage of *FvCKX6* transcripts through RLM-RACE in strawberry petals and GFP-Sensor tests in *N. benthamiana* leaves, indicating that miR159-mediated cleavage is a common mechanism for regulating abundance of *CKXs* (Supplementary Fig. 6).

Considering that GA-induced MYBs are conserved miR159 targets, we tested if the miR159–*GAMYB*s module contributes to cell division in petals. The three *GAMYB* members *RhMYB33*, *RhMYB65*, and *RhMYB101*

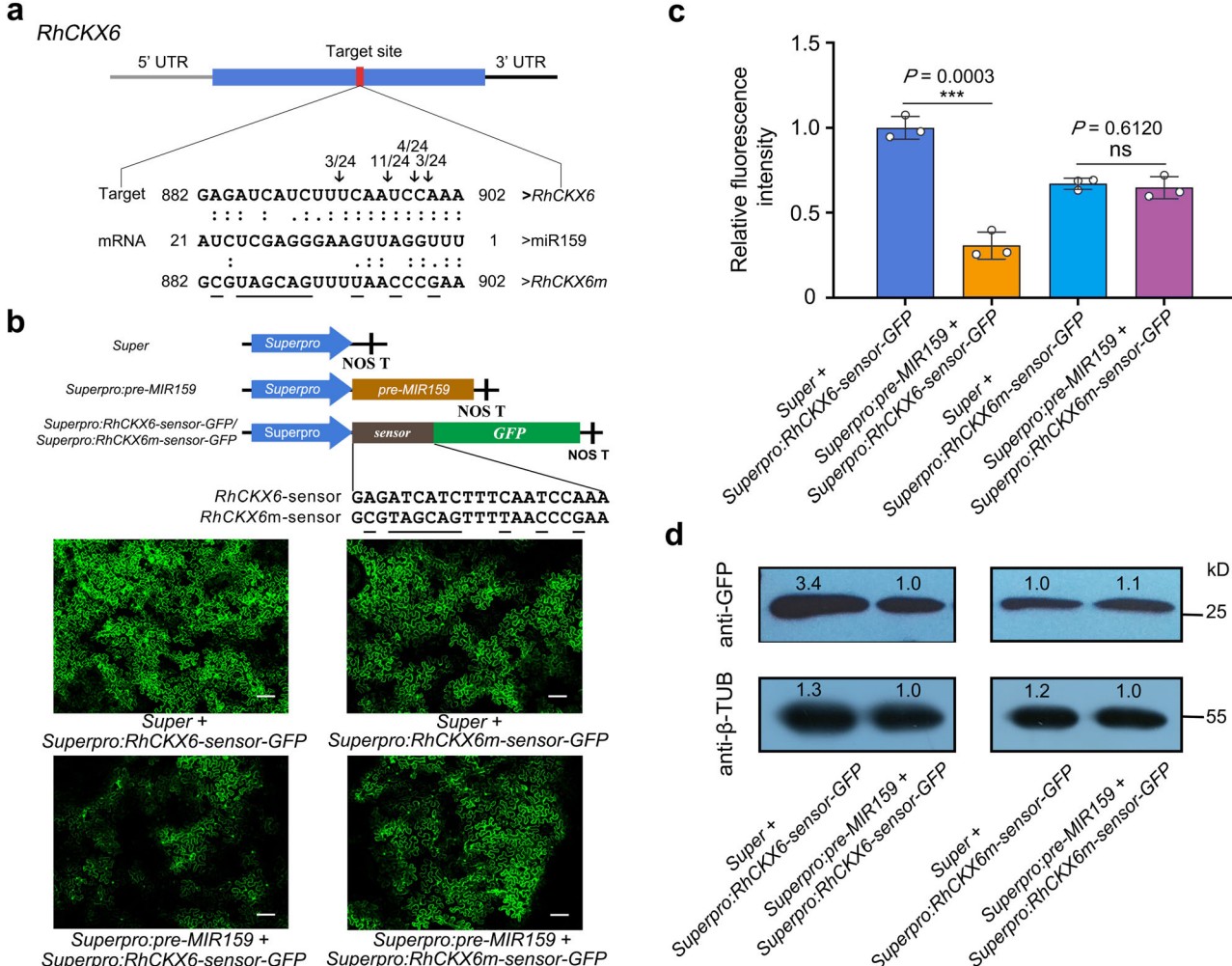

**Fig. 3 | miR159 targets *RhCKX6 in planta*. a** Validation of miR159-targeted cleavage of *RhCKX6*. Upper panel, diagram of *RhCKX6* mRNA. Red box, predicted cleavage site. Bottom panel, identification of cleavage sites using a 5′ RNA ligase-mediated rapid amplification of cDNA ends (5′ RLM-RACE) assay in rose petals. The positions of cleavage sites are indicated by arrows with the frequency of clones above. Dots represent Watson-Crick pairing. **b, c** Confocal imaging analysis (**b**) and relative fluorescence intensity (**c**) of *N. benthamiana* leaves 3 days after co-infiltrating *pre-MIR159* with *RhCKX6-sensor-GFP* or *RhCKX6m-sensor-GFP*. Upper panel, diagrams of the constructs overexpressing *pre-MIR159*, *RhCKX6-sensor-GFP*, and *RhCKX6m-sensor-GFP*. Bottom panel, GFP fluorescence in *N. benthamiana*

leaves 3 days after co-infiltrating the indicated constructs. Scale bars, 5 mm. The data in (**c**) are shown as means ± SD (*n* = 3). Asterisks indicate statistically significant differences (two-sided Student's *t*-test; ***$P < 0.001$; ns, no significant difference). **d** Immunoblot analysis of GFP protein levels in *N. benthamiana* leaves 3 days after co-infiltrating *pre-MIR159* with *RhCKX6-sensor-GFP* or *RhCKX6m-sensor-GFP*. Total proteins were extracted from leaves 3 days after infiltration and were subjected to immunoblot analysis using an anti-GFP antibody, with β-tubulin as an internal control. The experiments were performed independently three times, and representative results are shown (**b–d**).

were predicted as miR159 targets in rose (Supplementary Fig. 4). RNA-seq and RT-qPCR analyses failed to detect *RhMYB65* or *RhMYB101* expression in rose petals at all tested development stages (1 d, 3 d, and 5 d after stage 0). RLM-RACE in rose petals and GFP-Sensor tests in *N. benthamiana* leaves demonstrated the miR159-dependent cleavage of *RhMYB33* transcripts (Supplementary Fig. 7a, b). However, *RhMYB33* expression showed a slight but non-significant decrease in petals from 3 d to 7 d after stage 0 (Supplementary Fig. 7c), concomitant with a significant drop in miR159 levels (Fig. 2e). In addition, we measured GA levels in rose petals. GA₃ and GA₇ levels were very low, whereas GA₁ and GA₁₉ levels were stable throughout the developmental time course (Supplementary Fig. 7d).

miR159 and miR319 exhibit high sequence similarity and share overlapping target sites[63]. In Arabidopsis, miR319 targets the transcription factor gene *TCP4* to control floral organ morphology[40]. We thus tested whether miR319 is involved in regulating petal growth in rose. Small RNA blot and RT-qPCR analyses demonstrated that miR319 abundance is much lower than that of miR159, and remains

relatively constant during the entire petal growth period (Supplementary Fig. 8a–c). The expression levels of three predicted miR319 targets (*TCP2*, *TCP4*, and *TCP4-X1*) did not change in miR159-knockdown lines (Supplementary Fig. 8d). Based on these findings, we propose that the regulation of petal growth through the miR319–*TCP*s module is not as prominent as the regulation mediated by the miR159–*RhCKX6* module. However, we cannot rule out the possibility that miR319 and *TCP*s contribute to petal growth regulation in roses.

### The miR159–*RhCKX6* module regulates the duration of the cell division phase through modulating cytokinin catabolism in petals

To investigate the role of *RhCKX6* in petal growth, we silenced *RhCKX6* using a virus-induced gene silencing (VIGS) approach (Fig. 4a, b). We designed the TRV-*CKX6* construct using a gene-specific region of *RhCKX6* (Supplementary Fig. 9a). To test the specific silencing of *RhCKX6*, we performed RT-qPCR analysis: whereas *RhCKX6* levels

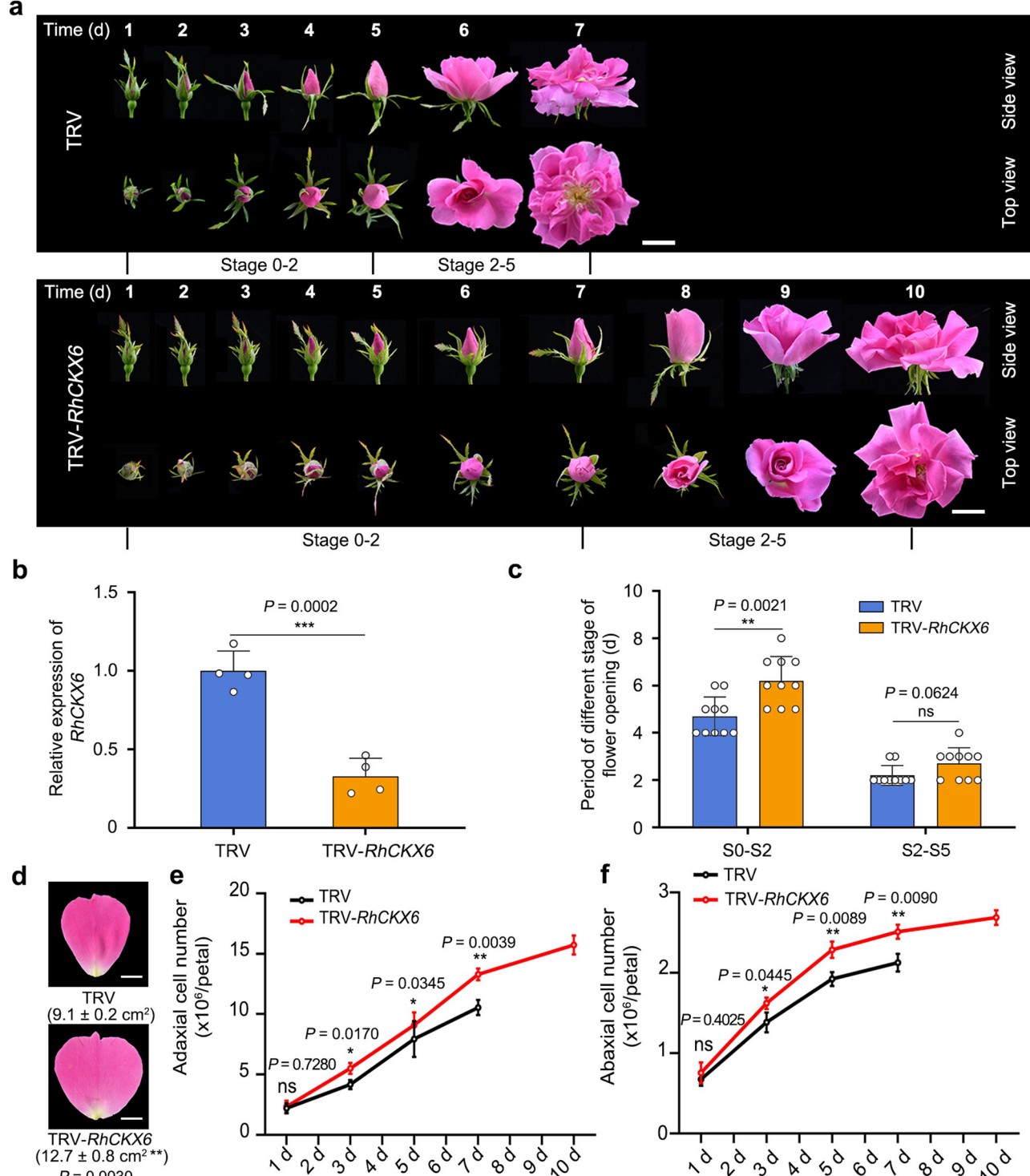

**Fig. 4 | *RhCKX6* influences the duration of the cell division phase and flower opening. a** Flower opening progression of TRV and TRV-*RhCKX6* plants. The experiments were performed independently twice with similar results, and one representative result is shown. Scale bars, 2 cm. **b** RT-qPCR analysis of *RhCKX6* transcript levels in sepals 5 days after stage 0. *RhUBI2* was used as an internal control. Data are shown as means ± SD ($n = 4$). **c** Flower development periods of TRV and TRV-*RhCKX6* plants. Data are shown as means ± SD ($n = 10$). **d** Size of petals in TRV and TRV-*RhCKX6* plants at the fully opened stage. The outermost layer of fully expanded petals was measured. Data are shown as means ± SD $n = 3$). Scale bars, 1 cm. **e, f** Number of adaxial epidermal (**e**) and abaxial epidermal (**f**) cells in TRV and TRV-*RhCKX6* petals from 1 day after floral stage 0 to the fully opened stage. Data are shown as means ± SD ($n = 3$). Asterisks indicate statistically significant differences (two-sided Student's *t*-test; *$P < 0.05$; **$P < 0.01$; ns no significant difference).

significantly decreased relative to the TRV controls, the expression levels of *RhCKX1* and *RhCKX5*, which have high sequence similarity to *RhCKX6*, were not significantly different between the *RhCKX6*-silenced lines and the TRV control (Supplementary Fig. 9b).

Compared to the TRV control, flower opening progression was delayed in *RhCKX6*-silenced lines due to a significantly extended period between stages 0 and 2 (S0–S2; 6.20 ± 0.98 days in *RhCKX6*-silenced plants and 4.70 ± 0.49 days in TRV) (Fig. 4c). In contrast to the

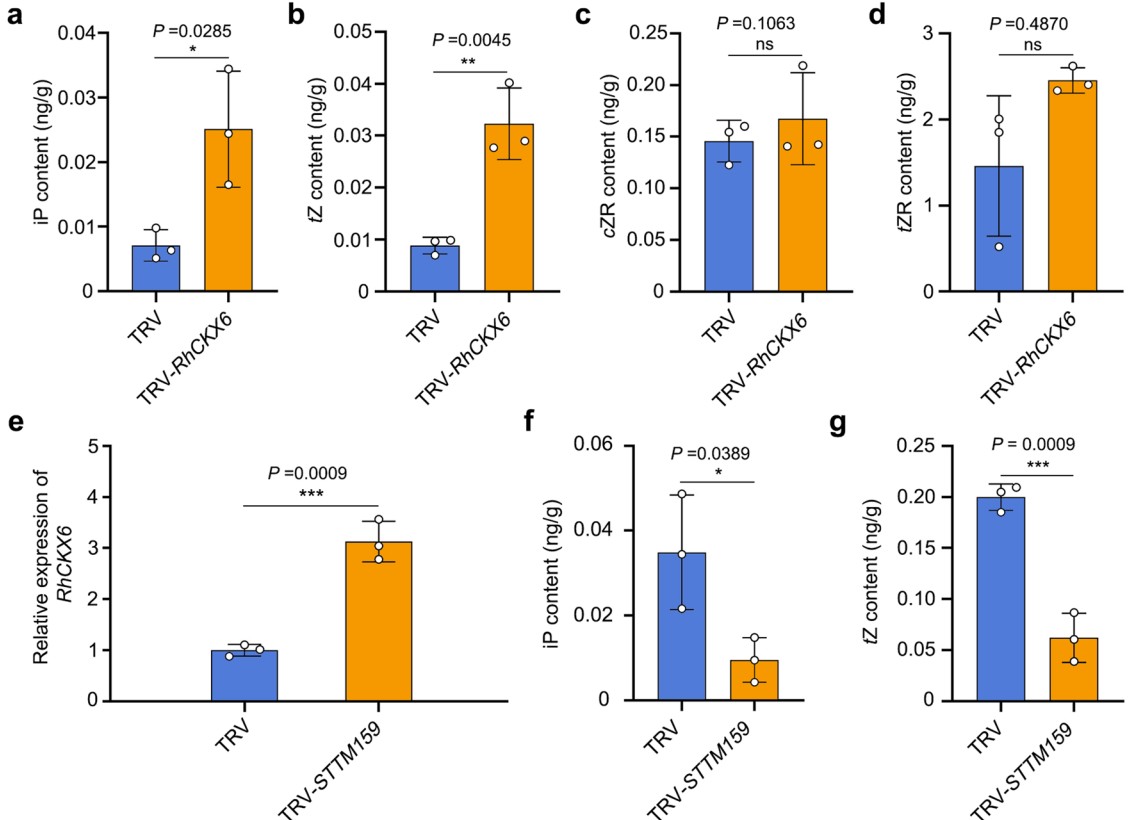

**Fig. 5 | miR159–*RhCKX6* module governs cytokinin catabolism in petals.**
**a**–**d** Cytokinin contents in petals of TRV and *RhCKX6*-silenced plants. Petals were sampled 5 days after stage 0. Data are shown as means ± SD (*n* = 3). **e** RT-qPCR analysis of *RhCKX6* transcript levels in petals of TRV and TRV-*STTM159* plants. *RhUBI2* was used as an internal control. Data are shown as means ± SD (*n* = 3).

**f**, **g** Cytokinin contents in petals of TRV and TRV-*STTM159* plants. Petals were sampled 3 days after stage 0. Data are shown as means ± SD (*n* = 3). Asterisks indicate statistically significant differences (two-sided Student's *t*-test; *$P$ < 0.05; **$P$ < 0.01; ***$P$ < 0.001; ns no significant difference).

miR159-knockdown lines, the final size of *RhCKX6*-silenced petals was larger than that of TRV controls (Fig. 4d). In agreement with this result, the numbers of adaxial and abaxial epidermal cells in *RhCKX6*-silenced petals were significantly higher than those of TRV controls (Fig. 4e, f). We also tested the effects of overexpressing pre-*MIR159* on rose petal size, as described previously[64]. Overexpressing pre-*MIR159* lowered *RhCKX6* transcript abundance, resulted in larger petals compared to control plants, and extended the cell division phase (from S0 to S2) (Supplementary Fig. S10). Based on these results, we conclude that the enhanced petal size resulting from *RhCKX6* silencing is most likely due to a prolonged cell proliferation phase.

Since CKX enzymes catalyze the irreversible degradation of cytokinins, we monitored cytokinin contents in petals of TRV control and *RhCKX6*-silenced lines[65–67]. The isopentenyladenine (iP) and *trans*-zeatin (tZ) levels in *RhCKX6*-silenced petals were significantly higher than those of the TRV controls, whereas the levels of *cis*-zeatin-riboside (*c*ZR), *trans*-zeatin-riboside (*t*ZR), and other cytokinin metabolites did not differ significantly (Fig. 5a–d; Supplementary Data 3). We monitored iP and *t*Z contents in petals of *STTM159* lines. Knocking down miR159 abundance was associated with significantly higher *RhCKX6* transcript levels and lower iP and *t*Z contents relative to the TRV controls (Fig. 5e–g).

Next, we determined the levels of iP and *t*Z in petals during early flower development (1 and 5 days after stage 0). The iP and *t*Z concentrations dropped significantly as petals grew, and reached very low levels 5 days after stage 0, which corresponds to the onset of cell expansion (Fig. 6a). We also tested whether an exogenous application of cytokinin (6-benzyladenine, 6-BA) effects petal growth and cell division duration. Indeed, 6-BA treatment significantly extended the

S0–S2 period (7.20 ± 0.98 days for the 6-BA treatment and 4.90 ± 0.74 days for the mock treatment) and increased the final petal size (Fig. 6b–d). The number of adaxial and abaxial epidermal cells was significantly higher in 6-BA-treated petals than in mock-treated petals, thus mimicking the phenotypes observed in *RhCKX6*-silenced lines (Fig. 4).

These results indicate that the miR159–*RhCKX6* module governs the progression of cell division by controlling the clearance of active cytokinins in petals during early petal development.

## A MYB–TPL–HDA19 complex regulates *MIR159* expression in petals by modulating H3K9ac levels at the *MIR159* promoter

To explore the regulatory mechanism of *MIR159* expression in rose petals, we isolated a 1500-bp *MIR159* promoter fragment. Promoter sequence analysis identified multiple binding sites for MYB transcription factors (Supplementary Fig. 11a). We generated *MIR159* promoter fragments (P1–P3) and tested their transactivation potential in rose when driving the expression of the *β-GLUCURONIDASE* (*GUS*) reporter gene. The P2 fragment (352–713 bp upstream from the transcription start site) exhibited the strongest transcriptional activity when transiently infiltrated in rose petals (Supplementary Fig. 11b). We therefore carried out a yeast one-hybrid (Y1H) assay using the P2 fragment to screen a cDNA library prepared from RNA extracted from mixed S0 and S2 rose petals. We obtained 39 candidate clones, including several encoding transcription factors (Supplementary Data 4). Further Y1H assays confirmed that only RchiOBHmChr2g0106671, a putative MYB transcription factor, binds to the P2 region of the *MIR159* promoter (Fig. 7a, Supplementary Fig. 11c). As RchiOBHmChr2g0106671 is closely related to Arabidopsis MYB73 (At4g37260) based on a

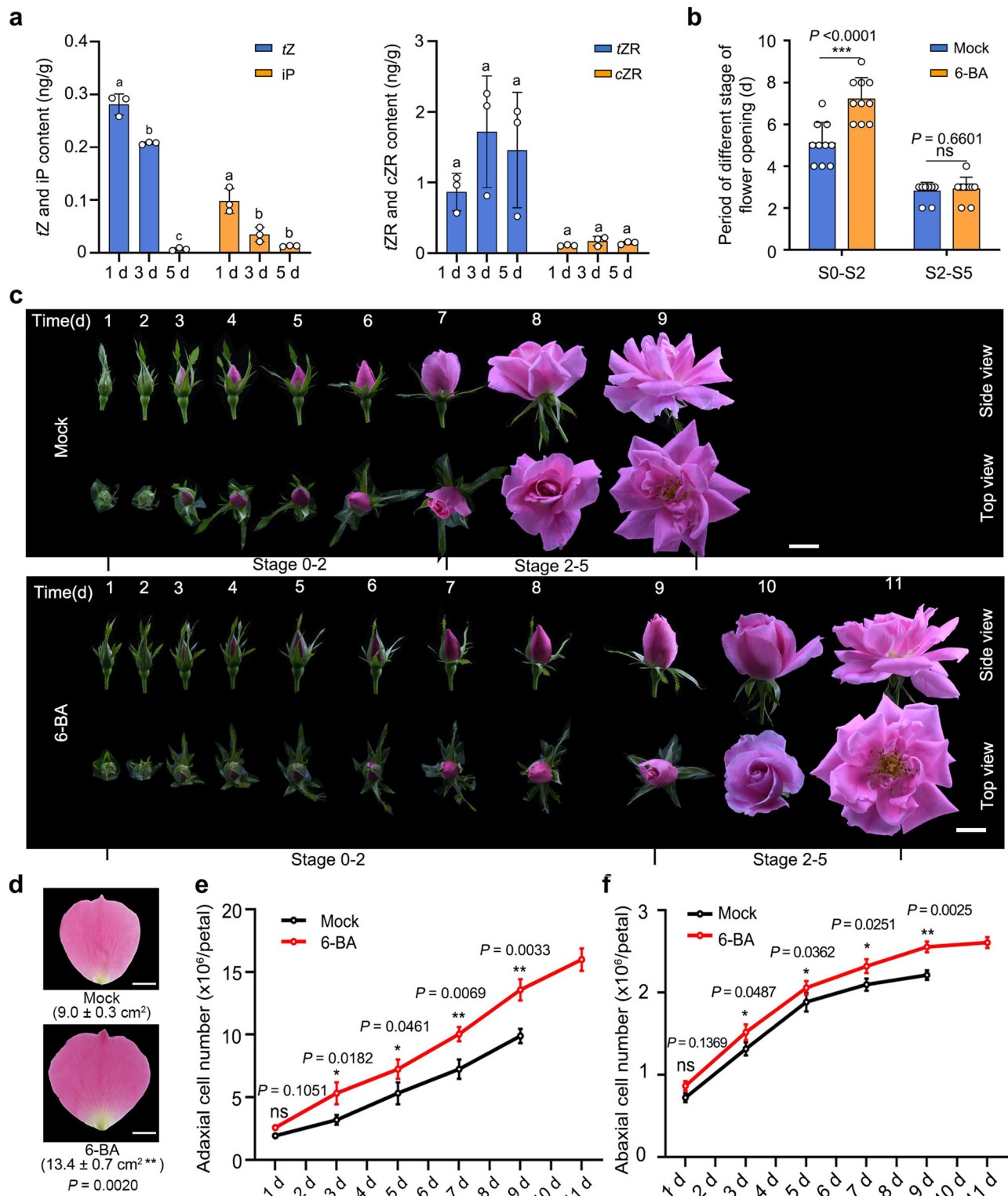

**Fig. 6 | Exogenous 6-benzylaminopurine (6-BA) prolongs the duration of the cell division phase in rose petals. a** Cytokinin contents in petals during the early development phase. The petals were sampled at 1 and 5 days after stage 0. Data are shown as means ± SD ($n = 3$). Different lowercase letters above each bar indicate significant differences according to one-way ANOVA with Tukey's multiple comparisons test ($P < 0.05$). **b, c** Flower opening progression of mock-treated and 6-BA-treated rose plants. For (**c**), Scale bars, 2 cm. Data are shown as means ± SD ($n = 10$).

**d** Petal size of mock-treated and 6-BA-treated rose at the fully opened stage. The outmost layer of fully expanded petals was measured. Data are shown as means ± SD ($n = 3$). Scale bars, 1 cm. **e, f** Number of adaxial epidermal (**e**) and abaxial epidermal (**f**) cells in mock-treated and 6-BA-treated petals from 1 day after floral stage 0 to the fully opened stage. Data are shown as means ± SD ($n = 3$). Asterisks indicate statistically significant differences (two-sided Student's $t$-test; *$P < 0.05$; **$P < 0.01$; ***$P < 0.001$; ns no significant difference).

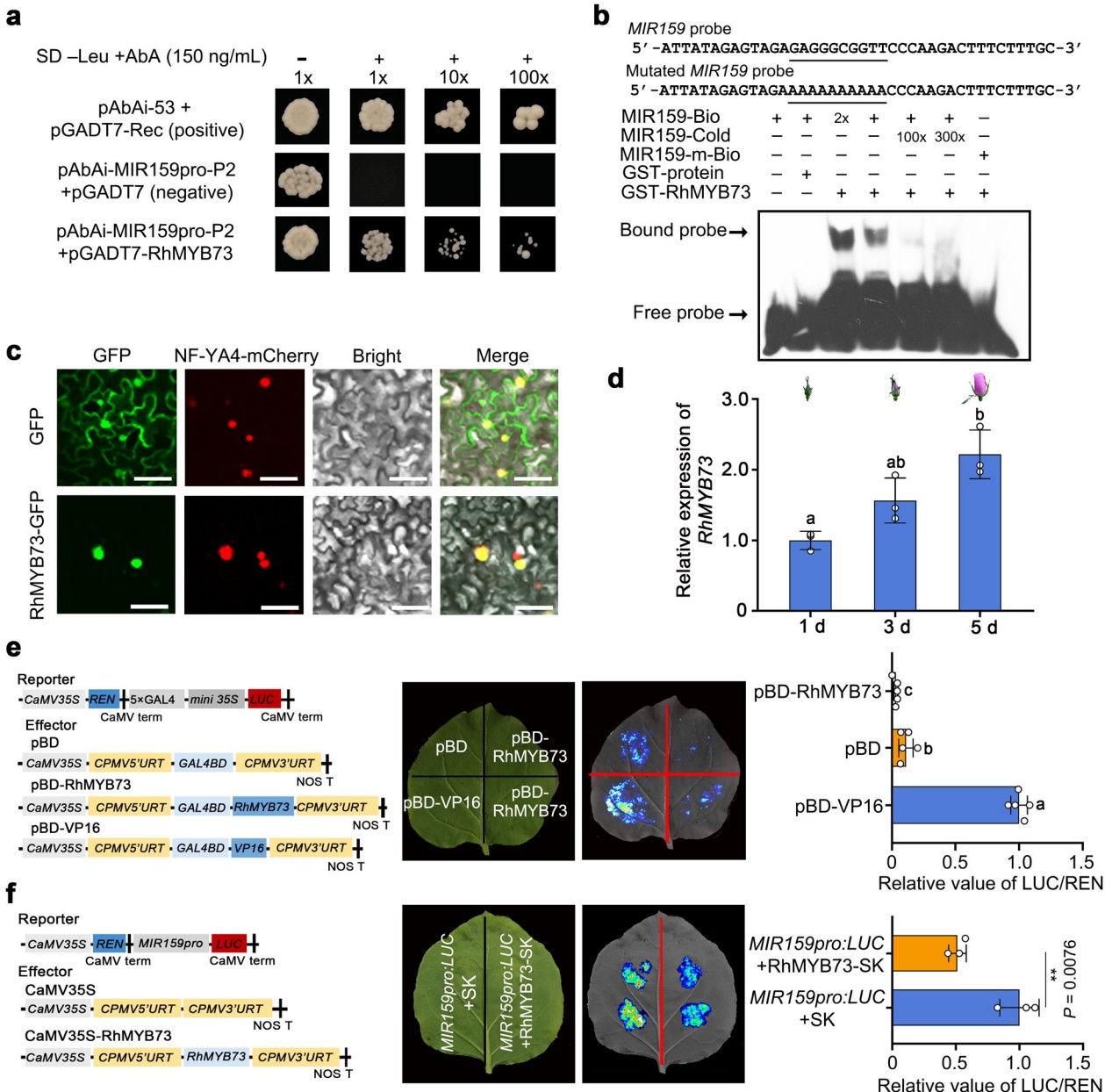

**Fig. 7 | RhMYB73 represses *MIR159* transcription. a** Yeast one-hybrid (Y1H) analysis of RhMYB73 binding to the *MIR159* promoter. "+" and "−" indicate synthetic defined (SD) medium lacking Leu with (+) or without (−) 150 ng mL⁻¹ Aureobasidin A (AbA). 1×, 10×, and 100× indicate the dilution factor of the yeast cultures before spotting onto the indicated medium (1, 10, and 100 times). **b** Electrophoretic mobility shift assay (EMSA) showing that RhMYB73 directly binds to the *MIR159* promoter. The region from −392 to −432 bp of the *MIR159* promoter was used to generate a probe. One microgram of purified recombinant GST-RhMYB73 was incubated with 2 nM biotin-labeled *MIR159* promoter probe. As indicated, RhMYB73-dependent mobility shifts were detected and competed by an unlabeled cold probe in a dose-dependent manner. **c** Subcellular localization of RhMYB73 in *N. benthamiana* leaves. *RhMYB73-GFP* was co-infiltrated with *NF-YA4-mCherry* (nuclear marker) into *N. benthamiana* leaves. Fluorescence signal was visualized by

confocal microscopy 3 days after infiltration. Scale bars, 50 μm. **d** RT-qPCR analysis of *RhMYB73* transcript levels in rose petals 1, 3, and 5 days after stage 0. *RhUBI2* was used as an internal control. Data are shown as means ± SD (*n* = 3). **e** Transcriptional repressor activity of RhMYB73 in *N. benthamiana* leaves. Left, diagrams of reporter and effector constructs. Middle images, brightfield and LUC activity. Right, quantitative analysis. Data are shown as means ± SD (*n* = 5). **f** Repression of transactivation of the *MIR159* promoter by RhMYB73. Left, diagrams of reporter and effector constructs. Middle images, brightfield and LUC activity. Right, quantitative analysis. Data are shown as means ± SD (*n* = 3). The experiments were performed independently twice with similar results, and one representative result is shown. Different lowercase letters (in (**d**) and (**e**)) indicate significant differences according to one-way ANOVA with Tukey's multiple comparisons test (*P* < 0.05). Asterisks (in (**f**)) indicate statistically significant differences (two-sided Student's *t*-test; **P < 0.01).

phylogenetic analysis, we named this protein RhMYB73 (Supplementary Fig. 12a). A search for conserved domains identified an R2R3 MYB domain in the N terminus and an ERF-associated amphiphilic repression (EAR) motif (LSLSL) in the C terminus (Supplementary Fig. 12b, c). An electrophoretic mobility shift assay (EMSA) confirmed the direct binding of RhMYB73 to the *MIR159* promoter in vitro (Fig. 7b). Subcellular localization suggested that RhMYB73 localizes to the nucleus

(Fig. 7c). We also determined that *RhMYB73* expression significantly increases from the S0 to S2 stages by RT-qPCR (Fig. 7d), which was opposite to the *MIR159* expression pattern. Transcriptional activity, as revealed by dual-luciferase activity assays, demonstrated that RhMYB73 is a transcriptional repressor (Fig. 7e). Indeed, co-infiltrating *RhMYB73* with the firefly luciferase gene (*LUC*) driven by the *MIR159* promoter resulted in lower LUC activity in planta compared to the

empty effector construct (Fig. 7f). Taken together, these results demonstrate that RhMYB73 directly binds to the *MIR159* promoter and represses its expression.

To further characterize the role of RhMYB73 in petal development, we silenced *RhMYB73* using VIGS. We chose a gene-specific region of *RhMYB73* to generate the TRV-*RhMYB73* construct (Supplementary Fig. 12d). RT-qPCR analysis showed that *RhMYB73* expression decreased (Fig. 8c), whereas that of *RhMYB70* and *RhMYB77*, which share high sequence similarity with *RhMYB73*, was not altered in *RhMYB73*-silenced lines, indicating that *RhMYB73* was specifically silenced (Supplementary Fig. 12e).

Notably, the earlier developmental window (stages S0–S2) of *RhMYB73*-silenced lines (5.60 ± 0.92 days) was significantly longer than that of the TRV controls (4.50 ± 0.67 days) (Fig. 8a–c). In contrast to the smaller petals of *STTM159* lines, the final size of *RhMYB73*-silenced petals was significantly larger than that of TRV controls (Fig. 8d), with more adaxial and abaxial epidermal cells (Fig. 8e, f). Compared to TRV controls, *RhMYB73* silencing caused a significant increase in miR159 abundance but lower *RhCKX6* transcript levels (Fig. 8g). In addition, the iP and *tZ* contents were also significantly higher in *RhMYB73*-silenced petals compared to the TRV controls (Fig. 8h). These results indicate that RhMYB73 regulates *MIR159* transcription and consequently the duration of the cell division phase in petals.

RhMYB73 harbors a conserved EAR motif, which has been well documented as a transcriptional repressor domain, mostly via recruiting chromatin remodeling factors[68-70]. We thus performed an immunoprecipitation-mass spectrometry (IP-MS) assay to identify possible RhMYB73-interacting partners and obtained 106 candidate proteins (Supplementary Data 5). Among them, we confirmed the interaction between RhMYB73 and the WD40-like family member RchiOBHmChr2g0134371 by yeast two-hybrid (Y2H) assay (Fig. 9a; Supplementary Fig. 13a). Phylogenetic analysis showed that RchiOBHmChr2g0134371 is closely related to Arabidopsis TOPLESS (TPL, At1g15750) and TOPLESS-RELATED1 (TPR1, At1g80490) (Supplementary Fig. 13b); thus, we named this protein RhTPL.

Mutating the EAR motif in RhMYB73 abolished its interaction with RhTPL, indicating that the EAR motif is indispensable for this interaction (Fig. 9a). The EAR motif acts as a scaffold for the TPL–Histone deacetylase (HDAC)–Polycomb repressive complex2 (PRC2) complex during petal organogenesis[71,72]. Therefore, we characterized the expression levels of *HDAC* genes during early petal development (stages 0–2). Of the 10 *HDAC* genes tested, *HDA6* (RchibHmChr7g0192801) and *HDA19* (RchiBHmChr6g0281731) expression substantially increased in petals during the transition from cell proliferation (stage 0) to cell expansion (stage 2) (Supplementary Fig. 13c, d). We conducted a bimolecular fluorescence complementation (BiFC) assay to test the interaction of RhTPL with RhHDA6 or RhHDA19 in *N. benthamiana* leaves co-infiltrated with the appropriate constructs and determined that RhTPL interacts with RhHDA19, but not with RhHDA6 (Fig. 9b; Supplementary Fig. 13e). Furthermore, BiFC assays showed that RhMYB73, RhTPL, and RhHDA19 interact with each other in nuclei in planta (Fig. 9b). A co-immunoprecipitation (co-IP) assay confirmed the RhMYB73–RhTPL and RhTPL–RhHDA19 interactions (Fig. 9c). RT-qPCR analysis indicated that *RhTPL* and *RhHDA19* expression increases significantly in petals during the progression from cell proliferation (stage 0) to cell expansion (stage 2) (Supplementary Fig. 13f, g).

HDA19 acetylates histone H3 on lysine 9 (K9) and K14 in plants[73]. Therefore, we monitored H3K9ac and H3K14ac levels at the *MIR159* promoter during early petal development (stages 0–2) by chromatin immunoprecipitation (ChIP)-qPCR. H3K9ac levels were significantly lower in fragments of the *MIR159* promoter recognized by RhMYB73 (P5 and P6) as petals progressed from S0 to S2 (Fig. 9d), whereas H3K14ac levels remained constant levels during the same developmental window (Supplementary Fig. 13h). Therefore, we monitored

H3K9ac levels of the *MIR159* promoter in TRV control and *RhMYB73*-silenced lines. As shown in Fig. 9e, compared to the TRV control, H3K9ac levels were significantly elevated at the P5 and P6 sites of the *MIR159* promoter in *RhMYB73*-silenced lines. Finally, a transactivation assay demonstrated that the presence of RhTPL and HDA19 significantly strengthens the repression of *MIR159* transcription imposed by RhMYB73 (Fig. 9f).

Taken together, we demonstrated that the MYB73–TPL–HDA19 repression complex inhibits *MIR159* transcription by modulating H3K9ac at its promoter. The resulting decrease in miR159 levels leads to *CKX6* transcript, and presumably CKX6 enzyme, accumulation, and thus precocious cytokinin clearance, consequently initiating an earlier transition to cell expansion in petals. This work describes a mechanism regulating the transition from cell proliferation to expansion that controls lateral organ size (Supplementary Fig. 14).

## Discussion

### Cytokinin accumulation is vital to lateral organ development through controlling the duration of the cell division phase

The final size of a given organ or tissue is dictated by the number of cells that comprise it and their size, which requires coordination between cell proliferation and expansion. The transition from cell proliferation to expansion is spatially and temporally regulated by a complex network that involves multiple phytohormones. Here, we discovered that a decrease in iP and *tZ* levels is associated with the onset of cell expansion in rose petals. It is important to note that lower cytokinin levels were achieved by triggering cytokinin catabolism through the accumulation of a cytokinin oxidase/dehydrogenase, RhCKX6. In agreement, exogenous application of cytokinin extended the duration of the cell division phase and delayed expansion-driven flower opening, suggesting that excess cytokinin impedes normal petal growth.

An earlier study integrating transcriptome profiling and image analysis showed that cytokinin-activating genes are rapidly down-regulated during the transition from proliferation to expansion in leaves, suggesting that cytokinins are involved in cell cycle arrest[10]. Moreover, cytokinins control the transition from the G1/S to G2/M phase during cell cycle progression by promoting the expression of *CycD3* and *Cyclin-dependent kinases*[74-76]. Consistent with this notion, a recent study showed that cytokinin levels peaked during vein and stomatal formation and then dropped sharply during cell expansion in leaves[18].

Cytokinin excess or deficiency also leads to abnormal leaf growth and smaller leaves[18,77]. Therefore, we propose that cytokinin homeostasis is vital for regulating the duration of the cell division phase. The temporally accurate clearance of cytokinins and the dampening of cytokinin responses are required for the normal development and growth of lateral organs and tissues, such as leaves and petals.

### miR159 regulates cytokinin catabolism by targeting *RhCKX6* transcripts

miR159 is an ancient and evolutionarily conserved miRNA that is present in all tested eudicot and monocot plants, as well as most angiosperms, gymnosperms, ferns, and lycopods[41]. The most conserved targets of miR159 encode GAMYB or GAMYB-like family proteins[42-46]. The miR159–GAMYB module influences multiple aspects of plant development[41,44]. In Arabidopsis, miR159 targets *MYB33* and *MYB65* to maintain normal leaf and flower development. Inhibiting miR159 function or mutating the miR159-binding site in *MYB33* and *MYB65* transcripts leads to developmental defects, such as stunted plants and curled leaves[46,78]. Conversely, overexpressing *MIR159* results in early flowering in Arabidopsis, rice (*Oryza sativa*), and wheat (*Triticum aestivum*) due to the downregulation of *GAMYB33* and *GAMYB65* transcript levels[79-81]. Moreover, the miR159–MYB33 module is involved in juvenile phase control since MYB33 directly binds to the

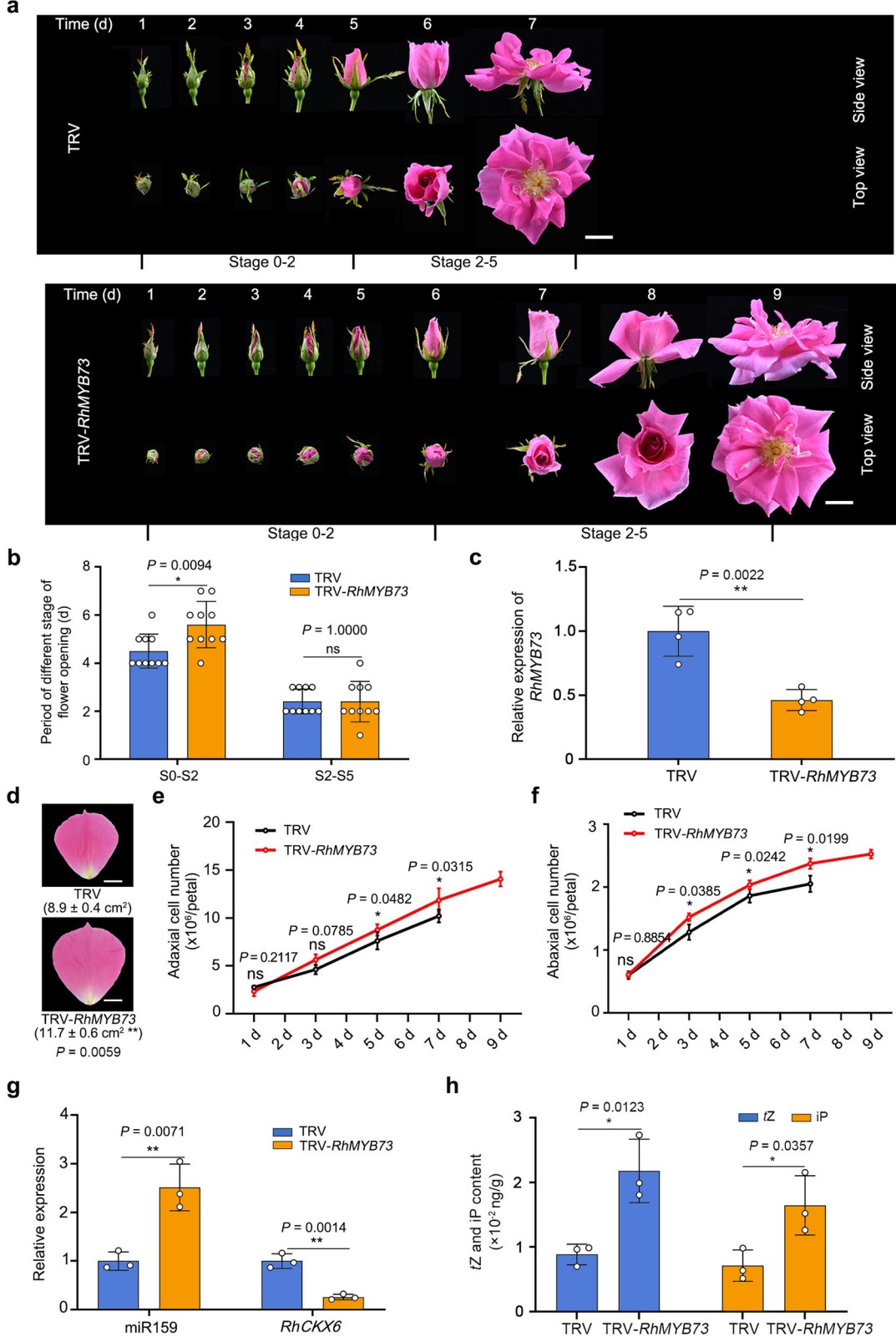

*MIR156* promoter and that of its other target gene *SQUAMOSA PRO-MOTER BINDING PROTEIN-LIKE9* (*SPL9*)[82]. In tomato, the miR159–*GA-MYB* module acts in ovules, and overexpressing *MIR159* leads to precocious fruit initiation and seedless fruits[83].

Besides these conserved *GAMYB* targets, about 20 other transcripts have been identified as miR159 targets in Arabidopsis according to bioinformatics analyses, several of which have been experimentally validated, including *1-AMINOCYCLOPROPANE-1-CARBOXYLATE SYN-THASE8* (*ACS8*, At4g37770) and *miRNA-REGULATED GENE1* (*MRG1*, At2g34010)[50,51].

In this study, we discovered that *RhCKX6* is a non-canonical target of miR159. We identified a miR159-binding site in the third exon of *RhCKX6* and detected authentic cleavage of *RhCKX6* transcripts in rose petals. Reduced miR159 abundance led to *RhCKX6* transcript

**Fig. 8 | Silencing *RhMYB73* prolongs the duration of the cell division phase in rose petals. a, b** Flower opening progression for TRV and TRV-*RhMYB73* plants. Data are shown as means ± SD (*n* = 10). Scale bars, 2 cm. **c** RT-qPCR analysis of *RhMYB73* transcript levels in sepals 5 days after stage 0. *RhUBI2* was used as an internal control. Data are shown as means ± SD from four biological replicates (*n* = 4). **d** Size of petals in TRV and TRV-*RhMYB73* plants at the fully opened stage. The outermost layer of fully expanded petals was measured. Data are shown as means ± SD (*n* = 3). Scale bars, 1 cm. **e, f** Number of adaxial epidermal (**e**) and abaxial epidermal (**f**) cells in TRV and TRV-*RhMYB73* petals from 1 day after floral stage 0 to the fully opened stage. Data are shown as means ± SD from three biological replicates (*n* = 3). **g** RT-qPCR analysis of miR159 abundance and *RhCKX6* transcript levels in petals of TRV and TRV-*RhMYB73* plants. *RhUBI2* was used as an internal control. Data are shown as means ± SD (*n* = 3). **h** Contents of *tZ* (left) and iP (right) in TRV and TRV-*RhMYB73* petals 5 days after stage 0. Data are shown as means ± SD (*n* = 3). The experiments were performed independently twice with similar results, and one representative result is shown. Asterisks indicate statistically significant differences (two-sided Student's *t*-test; **P* < 0.05; ***P* < 0.01; ns no significant difference).

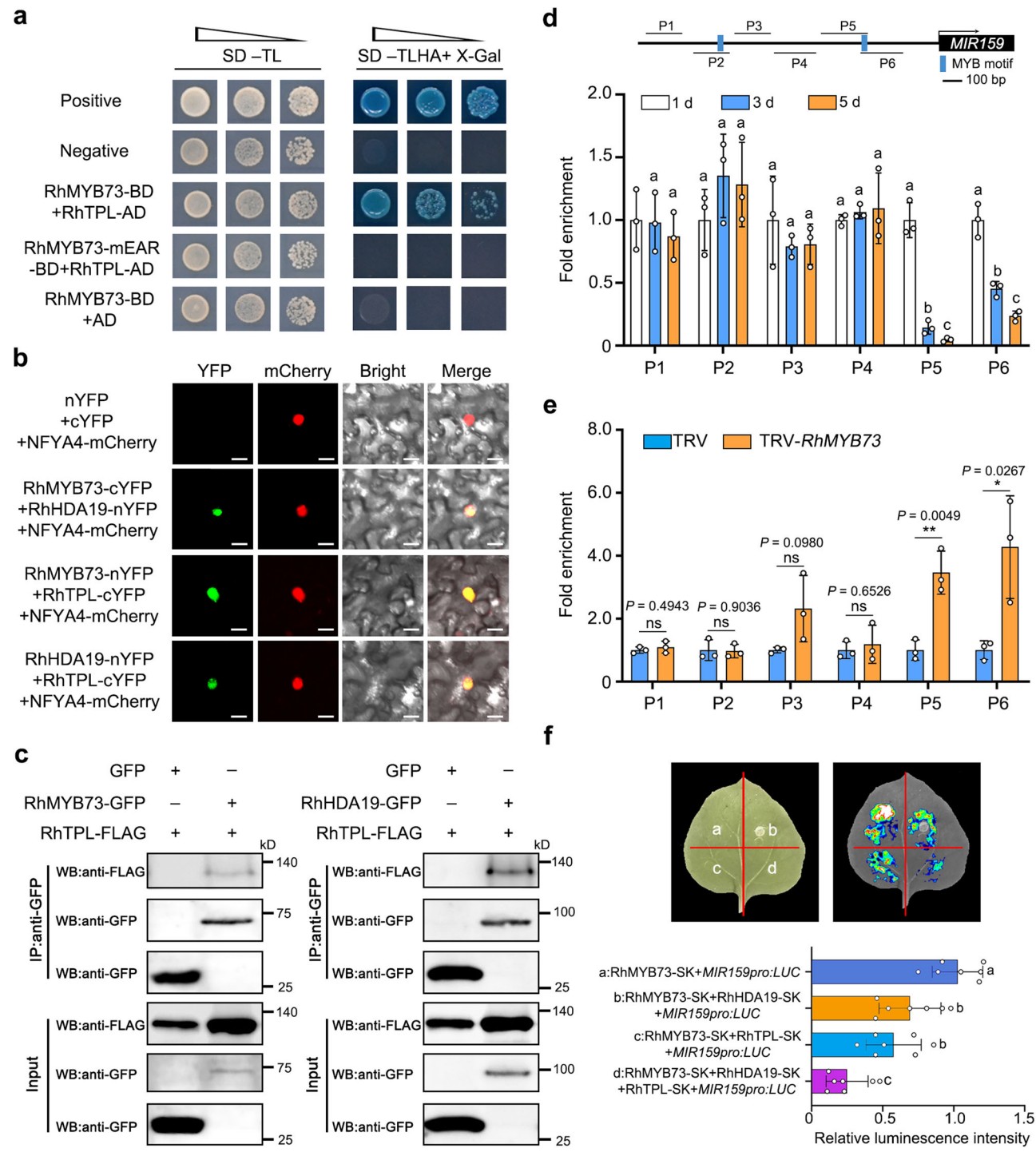

**Fig. 9 | RhMYB73 interacts with RhTPL and RhHDA19 to form a complex to repress *MIR159* transcription by modulating H3K9ac levels at the *MIR159* promoter. a** Yeast two-hybrid assay of the interaction between RhMYB73 and RhTPL. Synthetic defined (SD) medium −TL, SD −Trp −Leu; SD −TLHA + X-gal, SD −Trp −Leu −His −Ade + X-gal. Positive control, pGBKT7-53 + pGADT7-T; negative control, pGBKT7-lam + pGADT7-T. **b** Bimolecular fluorescence complementation assay testing the interaction between RhMYB73, RhHDA19, and RhTPL. *N. benthamiana* leaves were infiltrated with the construct combinations *RhMYB73-cYFP + RhHDA19-nYFP*, *RhMYB73-nYFP + RhTPL-cYFP*, and *RhTPL-cYFP + RhHDA19-nYFP*. The images were captured by confocal microscopy 3 days after infiltration. Scale bars, 20 μm. **c** Co-immunoprecipitation assay for the interaction between RhMYB73, RhHDA19, and RhTPL. Left, RhTPL and RhMYB73; right, RhTPL and RhHDA19. *RhTPL-FLAG* was co-infiltrated with *RhMYB73-GFP* or *RhHDA19-GFP* in *N. benthamiana* leaves. Total proteins were extracted 3 days after infiltration and the supernatant with soluble proteins was incubated with an anti-GFP antibody. The precipitates were analyzed by immunoblotting with anti-FLAG and anti-GFP antibodies. **d, e** Chromatin immunoprecipitation-quantitative PCR assay of the relative H3K9ac levels at the *MIR159* promoter in petals during early petal development (**d**) and in *RhMYB73*-silenced plants (**e**). Upper panel in (**d**), diagram of the RhMYB73-binding site in the *MIR159* promoter. For (**d**), the petals were sampled 1, 3, and 5 days after stage 0. Data are shown as means ± SD ($n = 3$). **f** Co-infiltration of RhHDA19 and/or RhTPL strengthens RhMYB73-mediated transcriptional suppression of *MIR159* in *N. benthamiana* leaves. Brightfield image (upper left), luciferase activity image (upper right), and quantitative analysis (bottom) of the transcriptional inhibition of the *MIR159* promoter by infiltration with the effector constructs *RhMYB73*, *RhMYB73 + RhHDA19*, *RhMYB73 + RhTPL*, and *RhMYB73 + RhHDA19 + RhTPL*. Data are shown as means ± SD ($n = 7$). The experiment was repeated independently three times, and one representative result is shown. Different lowercase letters (in (**d**) and (**f**)) indicate significant differences according to one-way ANOVA with Tukey's multiple comparisons test ($P < 0.05$). Asterisks (in (**e**)) indicate statistically significant differences (two-sided Student's *t*-test; *$P < 0.05$; **$P < 0.01$; ns no significant difference).

accumulation and cytokinin catabolism, which consequently stopped cell division. Notably, *RhCKX6* was recently reported to regulate petal dehydration tolerance and senescence, similar to a gene downstream of the NAC transcription factor RhNAP, suggesting that *RhCKX6* plays multiple roles in petal growth and in the response to abiotic stress[84].

Notably, the expression of two predicted canonical *GAMYB* targets of miR159, *RhMYB65* and *RhMYB101*, was barely detectable in rose petals during petal growth. The expression of *RhMYB33*, another *GAMYB*, remained relatively constant and declined slightly in fully opened petals. Unlike cytokinins, GA levels did not exhibit significant changes in petals. GA₃ and GA₇ levels were hardly detectable, whereas $GA_1$ and $GA_{19}$ levels remained stable throughout the tested developmental timeline in the petals. It appears that GA-responsive *GAMYB*s do not contribute to petal cell division in rose.

miR159 and miR319 exhibit high sequence similarity and share overlapping target sites[63]. In Arabidopsis, miR319 targets transcription factor genes from the *TCP* family to control floral organ morphology. Expressing a miR319-resistant form of *TCP4* under the control of the petal- and stamen-specific *APETALA3* (*AP3*) promoter led to a complete absence of petals and stamens[40], indicating that the miR319−*TCP4* module is involved in petal and stamen primordia initiation. In rose petals, small RNA blot and RT-qPCR analyses demonstrated that miR319 abundance is much lower than that of miR159 and remains relatively constant during petal growth. Furthermore, the expression levels of three predicted miR319 targets (*TCP2*, *TCP4*, and *TCP4-X1*) did not change in miR159-knockdown lines. We thus propose that the miR319−*TCP*s module plays a modest role in regulating petal growth compared to the miR159−*RhCKX6* module in roses. However, it is still possible that the miR319−*TCP*s module contributes to petal growth regulation and further investigation is required to verify its involvement.

In conclusion, the miR159−RhCKX6 module represents a regulatory mechanism controlling the duration of the cell division phase in petal growth.

### *MIR159* expression is regulated by an epigenetic suppression complex

Here, we determined that *MIR159* expression is developmentally regulated by a repressive epigenetic complex. The R2R3 MYB transcription factor RhMYB73 directly bound to the *MIR159* promoter. RhMYB73 harbors an EAR motif and recruited RhTPL, which in turn interacted with RhHDA19. This MYB73−TPL−HDA19 complex modulated H3K9ac levels at the *MIR159* promoter to determine the transcriptional output of *MIR159*, consequently influencing cytokinin catabolism in petals.

In Arabidopsis leaves, cytokinin responses are also controlled by a chromatin remodeling complex. The CIN-TCP transcription factor TCP4 interacts with the Switch/Sucrose Non-Fermentable (SWI/SNF)

ATPase BRAHMA (BRM) and thus activates *ARABIDOPSIS RESPONSE REGULATOR16* (*ARR16*) and *ARR6* expression by modifying the chromatin state of their promoters. As negative regulators of the cytokinin response, higher levels of *ARR16* and *ARR6* reduce leaf sensitivity to cytokinins and promote cell expansion and leaf maturation[77]. Therefore, cytokinin homeostasis (in this study) and response are tightly controlled by epigenetic mechanisms during the transition from cell proliferation to cell expansion, indicating that cytokinin levels are crucial for proper lateral organ development.

## Methods

### Plant materials and growth conditions
Rose (*Rosa hybrida* cv. 'Samantha') plantlets were harvested from commercial greenhouses for propagation[85]. Explants at the one-node-stem were cultured on propagation medium for 30 days and then transferred to rooting medium for 30 days[86]. The rooted plants were transplanted into pots (9-cm diameter) containing peat soil:vermiculite (1:1, v/v) and grown at $22 \pm 1\,°C$, with 50% relative humidity, under a 16-h-light/8-h-dark photoperiod and $100\,\mu mol\,m^{-2}\,s^{-1}$ illumination with fluorescent lamps (SINOL, SN-T5, and 16 W). *Nicotiana benthamiana* plants were grown in the same conditions.

### Virus-induced gene silencing (VIGS)
VIGS was performed as described in Zhang et al.[85]. Gene-specific fragments for *RhMYB* (402 bp), *STTM159* (short tandem target mimic of miR159, 96 bp), and *RhCKX6* (408 bp) containing *Eco*RI and *Kpn*I restriction sites at either end were used to construct the plasmids pTRV2-*RhMYB73*, pTRV2-*STTM159*, and pTRV2-*RhCKX6*, respectively. Agrobacterium (*Agrobacterium tumefaciens*) strain GV3101 individually carrying pTRV2-*RhMYB73*, pTRV2-*STTM159*, or pTRV2-*RhCKX6* was grown in LB medium containing $50\,mg\,L^{-1}$ kanamycin and $50\,mg\,L^{-1}$ rifampicin and shaken at 200 rpm at 28 °C for 16 h. Agrobacterium cells were collected by centrifugation at 5000 rpm for 8 min at room temperature and resuspended in infiltration buffer (10 mM $MgCl_2$, 200 mM acetosyringone, and 10 mM 2-(*N*-Morpholino)-ethanesulfonic acid, pH 5.6) to a final $OD_{600} = 1.5$. The cultures of pTRV1 and pTRV2-*RhMYB73*, pTRV2-*STTM159*, and pTRV2-*RhCKX6* were mixed in a 1:1 (v/v) ratio and incubated in the dark at room temperature for 3–4 h. Entire rose plants were immersed in infiltration buffer and exposed to −25 kPa vacuum for 10 min, twice. The infiltrated plants were briefly washed with deionized water and transplanted to pots containing peat soil:vermiculite (1:1, v/v) and grown at $22 \pm 1°C$, with 50% relative humidity, under a 16-h-light/8-h-dark photoperiod. Before analysis, *RhMYB73*-silenced, *STTM159* knockdown, and *RhCKX6*-silenced plants were identified by determining the transcript levels of the respective gene (*RhMYB73*, *STTM159*, or *RhCKX6*) by RT-qPCR in sepals. According to our previous research, phenotypes of early flower development were monitored and analyzed when floral buds reached stage 0 (bud

revealed)[56,59,60,87]. The number of plants displaying the selected phenotypes for each VIGS line TRV2-*STTM159*, *RhCKX6*, and *RhMYB73* were 61/116, 53/126, and 57/122, respectively. The primers used in the VIGS assay are listed in Supplementary Data 6.

### 5′ RNA ligase-mediated rapid amplification of cDNA ends (5′ RLM-RACE)

The 5′ RLM-RACE assay was performed according to the manual of the FirstChoice RLM-RACE kit (No. AM1700). Total RNA was extracted with a pBIOZOL kit (BioFlux, BSC55Ml) from rose and wild strawberry petals, treated with calf intestine alkaline phosphatase (CIP) and tobacco acid pyrophosphatase, and ligated with RNA adapters at 37 °C for 1 h. Reverse transcription and nested PCR for 5′ RLM-RACE (outer and inner) PCR were performed according to the manufacturer's instructions. The second PCR products were ligated into the pEASY-Blunt Cloning Vector (No. CB111; Transgene, Beijing, China) for sequencing. Primers used in RLM-RACE are listed in Supplementary Data 6.

### Quantification of endogenous cytokinins

About 300 mg (fresh weight) of rose petals were harvested, immediately frozen in liquid nitrogen, and stored at −70 °C. The cytokinin contents were detected by MetWare (Wuhan, China) based on the AB Sciex QTRAP 6500 LC-MS/MS platform (UPLC, ExionLC AD, https://sciex.com.cn/; MS, Applied Biosystems 6500; Triple Quadrupole, https://sciex.com.cn/). Three biological replicates were performed for each sample.

### Exogenous 6-benzylaminopurine (6-BA) treatment

The 6-BA treatment was performed following the procedure described previously[88]. Flower buds (bud revealed but ahead of stage 0) were sprayed with 100 μM 6-BA every day until 5 days after stage 0.

### Chromatin immunoprecipitation-quantitative PCR (ChIP-qPCR)

The ChIP-qPCR was performed as described in Chen et al.[87]. Briefly, 1.5 g of rose petals (collected at 1, 3, and 5 days after floral buds reached stage 0) was crosslinked in 1% (w/v) polyformaldehyde. Chromatin was fragmented into >500-bp segments via sonication, followed by incubation and immunoprecipitation with a specific antibody against acetylated histone H3K9/K14 (H3K9ac, 1:5000 dilution, Abcam, No. ab10812; H3K14ac, 1:5000 dilution, Millipore, No. 07-353). The immunoprecipitated DNA fragments were analyzed by quantitative PCR. The primers used in ChIP-qPCR are listed in Supplementary Data 6.

### Small RNA blot analysis of miRNA

Total RNA was extracted from rose petals with CTAB reagent. Small RNA blotting was performed following the protocol described in Lee et al. and Kim et al.[89,90]. Briefly, 20 μg RNA was separated on a 15% urea-polyacrylamide gel and transferred to a nylon membrane (Millipore, No. INYC00010) using a semidry transfer cell (Bio-Rad, Parameter: 390 mA, 20–30 min). Subsequently, hybridization was performed according to a standard protocol. Locked Nucleic Acids (LNA)-modified oligonucleotide probes were used to ensure the specificity of detection for miR159 and miR319[91]. The probes corresponding to miR159 and miR319 were 5′-taGAGCTCCCTTCAATCCAAa-3′ and 5′-ggAGCTCCCTTCAGTCCAAt-3′, respectively (lowercase letters indicate the LNA-modified oligonucleotides). U6 (5′-AGGGGCCATGC-TAATCTTCTC-3′) was used as an internal control.

### Stem-loop RT-qPCR for miRNA

Total RNA was extracted from rose petals with CTAB reagent. Reverse transcription was performed with 200 ng total RNA with a miRNA-specific stem-loop RT primer using Takara PrimeScript RT reagent kit (RR037A). qPCR was performed according to a previously described

method using a miRNA-specific forward primer and a universal reverse primer[92]. 5S rRNA was used as an internal control.

### Microscopy observation and cell counting

Cell imaging and cell counting were performed as described by Ma et al. and Yamada et al.[55,56]. The outermost petals were peeled off and fixed by formaldehyde-acetic acid (FAA) solution (3.7% formaldehyde, 5% glacial acetic acid, 50% ethanol) and decolorized with an ethanol gradient series. The adaxial and abaxial epidermal cells of the petals were observed with an optical microscope and photographed. The number of cells was counted using Image J (National Institutes of Health, USA) per visual field. The total number of adaxial and abaxial epidermal cells per petal was calculated by dividing the total petal area by the number of cells per unit area.

### RNA extraction and quantitative RT-PCR (RT-qPCR)

Total RNA was extracted from rose petals by a pBIOZOL kit (BioFlux, BSC55Ml)[88]. Genomic DNA contaminants were removed by digestion with RNase-free DNase I (Promega). Reverse transcription was performed with 1 μg total RNA using HiScript II Q Select RT SuperMix kit (Vazyme, Nanjing, China). qPCR was performed on a StepOne Real-Time PCR System (Applied Biosystems, Carlsbad, USA) and a KAPA SYBR FAST qPCR kit (Kapa Biosystems, Wilmington, MA, USA). *RhUBI2* was used as an internal control[86].

### RNA-seq

Total RNA was extracted from rose petals as described in Wu et al. and Tian et al.[88,93], and genomic DNA contamination was removed using RNase-free DNase I (Promega). The quality and quantity of the RNA were assessed using a NanoPhotometer® spectrophotometer (IMPLEN, CA, USA) and an Agilent 2100 Bioanalyzer (Agilent Technologies, CA, USA). Samples with an RNA integrity number (RIN) > 8 were used for RNA-seq by Beijing Novogene Bioinformatics Technology Co., Ltd. (Beijing, China). One microgram of total RNA per sample was used for cDNA library construction and Illumina sequencing, and RNA-seq data were processed, assembled, and annotated as previously described[94,95]. Clean RNA-seq reads were aligned to the reference genome (*Rosa chinensis* cv. Old Blush, GenBank ID 8255808) and deposited in the NCBI BioProject database under the accession number PRJNA808873. Finally, differentially expressed genes (DEGs) were analyzed according to the parameters (fold change ≥ 2 or ≤0.5, $q$-value ≤ 0.05) and subjected to GO and KEGG enrichment analyses. The stage 0 and stage 2 microarray datasets were obtained from the rose transcriptome database (http://bioinfo.bti.cornell.edu/rose).

### Subcellular localization

The coding sequence of *RhMYB73* was cloned in-frame of the *GFP* sequence into the pSuper vector containing *Xma* I and *Kpn* I restriction sites. pSuper:*NF-YA4*-mCherry was used as a nuclear marker[85]. The constructed vectors were transformed into Agrobacterium strain GV3101 and co-infiltrated into *N. benthamiana* leaves for 3 days. Subcellular localization was monitored in leaf epidermal cells by confocal laser scanning microscopy (Olympus, FV3000, Japan) at 488 nm and 561 nm excitation, respectively. The primers used in the assay are listed in Supplementary Data 6.

### GUS staining and yeast one-hybrid assay (Y1H)

A 1500-bp fragment of the *MIR159* promoter was cloned based on the rose genome sequence (https://lipm-browsers.toulouse.inra.fr/pub/RchiOBHm-V2/) with primers using genomic DNA from the cultivar 'Samantha' as template. The cloned *MIR159* promoter (−1000 to −1 bp relative to the transcription start site of *MIR159*) was truncated into three short segments (P1, P2, and P3) and then cloned into a modified *35Smini:GUS* vector with a minimal 35S promoter upstream of the *GUS* reporter gene. GUS staining was performed 3 days after infiltration of

rose petals through Agrobacterium-mediated assays. After analysis of *MIR159* promoter activity, the *MIR159* promoter fragment with highest GUS activity and flanked by *Hind* III and *Sal* I restriction sites was inserted into the pAbAi vector to generate the reporter construct pAbAi-*proMIR159*-P2. In parallel, the coding sequence of *RhMYB73* flanked by *EcoR* I and *Xho* I restriction sites was inserted into the pGADT7 vector to generate the effector construct pGADT7-RhMYB73. The Y1H assay was performed to examine the interaction between RhMYB73 and the *MIR159* promoter according to manual provided by Matchmaker Gold Y1H Library Screening System (Clontech, Dalian, China). The resulting vectors were transformed into yeast strain Y1H gold, and the transformants were grown for 3 days on synthetic defined (SD) medium lacking Ura and Leu (SD −Ura −Leu) with or without 150 ng mL$^{-1}$ Aureobasidin A (AbA). Primers used for Y1H are listed in Supplementary Data 6.

### Electrophoretic mobility shift assay (EMSA)
The EMSA was performed as described in Pei et al. and Cheng et al.[60,96]. The *RhMYB73* coding sequence was inserted into the pGEX-4T-2 vector at the *EcoR* I and *Xho* I restriction sites to generate pGEX-4T-RhMYB73. The resulting vector was transformed into *Escherichia coli* strain BL21 to produce recombinant GST-RhMYB73 fusion protein, which was induced by the addition of isopropyl-β-D-thiogalactopyranoside (IPTG, 0.2 mM) at 28 °C for 6 h. The GST-RhMYB73 protein was purified with glutathione Sepharose 4B (GE Healthcare, Pittsburgh, USA) following the manufacturer's instructions. The *MIR159* promoter probe was synthesized and labeled with biotin. Unlabeled and mutated biotin probes for the *MIR159* promoter were used as competitors. EMSAs were performed using a chemiluminescent nucleic acid detection module kit (Thermo Fisher, Waltham, USA). The primers used in EMSA are listed in Supplementary Data 6.

### Dual-LUC reporter assay
The dual-LUC reporter assay was performed as described in Zhang et al.[86]. For transcriptional activity analysis, the coding sequence of *RhMYB73* was inserted into the pBD vector[97]. The transcriptional activity of RhMYB73 was compared to the positive (pBD-VP16) and negative (pBD) controls[98]. For the transcriptional inhibition of *MIR159* transcription by RhMYB73, the coding sequence of *RhMYB73* was inserted into the pGreenII 62-SK vector to construct the *35S:RhMYB73* effector plasmid, and the *MIR159* promoter with 1000 bp in length was inserted into pGreenII 0800-LUC to construct the *MIR159pro:LUC* reporter plasmid. The resulting vectors were individually transformed into Agrobacterium strain GV3101 (harboring the pSoup plasmid) and co-infiltrated into *N. benthamiana* leaves. Three days later, we cut the tobacco leaves that have been injected with *Agrobacterium* and sprayed a reaction solution containing 50 mg L$^{-1}$ of luminol substrate D-luciferin onto the abaxial surface of the leaves. The reaction was carried out for 5 min in the dark, and then luciferase activity images were captured with a CCD camera (CHEMIPROHT 1300B/LND, 16 bit; Roper Scientific, Sarasota, FL, USA) to detect luminescence, with an exposure frequency of 100 KHz for 10 min. The ratios between firefly (LUC) and *Renilla* (REN) luciferase activities were measured using dual-LUC assay reagents (Promega, Madison, WI, USA). Primers used for the LUC assay are listed in Supplementary Data 6.

### Phylogenetic analysis
Alignment was performed with ClustalW (https://www.genome.jp/tools-bin/clustalw). The phylogenetic tree was constructed based on the alignment results using the maximum likelihood algorithm in MEGA version X software with 1000 bootstrap replicates.

### Yeast two-hybrid assay (Y2H)
The Y2H was performed as described in Chen et al.[87]. The coding sequences of *RhMYB73* and *RhMYB73*-mEAR (with a mutated EAR

sequence) were inserted into pGBKT7 (BD) to construct RhMYB73-BD as bait vector. The coding sequence of *RhTPL* was inserted into pGADT7 (AD) to construct RhTPL-AD as prey vector. The bait and prey vectors, positive control vectors (pGBKT7-53 and pGADT7-T), and negative control vectors (pGBKT7-lam and pGADT7-T) were transformed into yeast strain Y2H Gold. The positive colonies were transferred to different growth media (SD −TL, SD −Trp −Leu; SD −TLHA, SD −Trp−Leu−His−Ade; SD −TLHA X-gal SD −Trp−Leu−His−Ade + X-gal). The primers used in Y2H are listed in Supplementary Data 6.

### Bimolecular fluorescence complementation (BiFC)
The BiFC was performed as described in Chen et al.[87]. The coding sequences of *RhMYB73*, *RhHDA19*, and *RhTPL* were cloned in-frame with the sequence encoding the N-terminal half of YFP (nYFP) or the C-terminal half of YFP (cYFP) to generate *RhMYB73-nYFP*, *RhMYB73-cYFP*, *RhHDA19-nYFP*, and *RhTPL-cYFP* vectors. The appropriate pairs of constructs (*RhMYB73-nYFP* and *RhTPL-cYFP*; *RhMYB73-cYFP* and *RhHDA19-nYFP*; *RhHDA19-nYFP* and *RhTPL-cYFP*) were infiltrated into *N. benthamiana* leaves via *Agrobacterium*-mediated transient infiltration. At 3 days after infiltration, YFP and mCherry fluorescence was detected by confocal microscopy (Olympus, FV3000, Japan) with excitation at 488 and 561 nm, respectively.

### Immunoprecipitation-mass spectrometry (IP-MS)
For IP-MS assay, total proteins were extracted using a Total Protein Extraction Kit for Plant Tissues (Minute™, inventbiotech, Beijing, China) and incubated with recombinant GST or GST-RhMYB73 for 2 h at 4 °C. Supernatant was transferred to a fresh tube and purified with glutathione Sepharose 4B (GE Healthcare, Pittsburgh, USA) following the manufacturer's instructions. The samples were collected with centrifugation at 5000 rpm at 4 °C for 10 min. The purified proteins were fractionated on 10% (v/v) SDS-PAGE gels and digested with trypsin overnight incubation at 37 °C for subsequent MS analyses. The resulting MS/MS data were processed using the MaxQuant search engine (v.1.5.2.8)[99]. The instrument was equipped with an electrospray ionization-trap (ESI-TRAP). The search parameters were as follows: mass tolerance was set to 20 parts per million (ppm) for the search. The fragment mass tolerance was ±0.02 Da. Trp-C/M was specified as the cleavage enzyme while allowing up to two missing cleavages. The raw mass spectrometry data of validation have been deposited to the ProteomeXchange Consortium under accession number PXD046197. The primers used in IP-MS are listed in Supplementary Data 6.

### Co-immunoprecipitation assays (Co-IP)
For Co-IP assays, Agrobacterium harboring *35S:RhHDA19-GFP* was co-infiltrated with *35S:RhTPL-FLAG* into *N. benthamiana* leaves, *35S:RhMYB73-GFP* was co-infiltrated with *35S:RhTPL-FLAG* into *N. benthamiana* leaves, and co-infiltration of *35S:GFP* and *35S:RhTPL-FLAG* was used as a negative control. After 3 days, total proteins were extracted using extraction buffer[87]. The supernatant was first incubated with GFP-Trap® Agarose Beads (1:200 dilution; Chromotek, No. gta) and then analyzed by immunoblot using anti-FLAG (1:2500 dilution; Abcam, No. ab1162,) and anti-GFP (1:2000 dilution; Sigma-Aldrich, No. G1544,) antibodies. The primers used in Co-IP are listed in Supplementary Data 6.

### Transient overexpression in rose bud
The pSuper::*pre-MIR159* construct and pSuper were transformed into Agrobacterium tumefaciens cells (strain GV3101).The rose bud (0.5 cm) was immersed in the pSuper::*pre-MIR159* and pSuper bacterial suspension for transient overexpression of *MIR159*, followed by infiltrating under a vacuum at 0.7 MPa[64]. Before analysis, vector primers pSuper-F/R were used to measure DNA level insertions in petals at the flower opening stage 2 of inoculated plants by PCR. Then, we identified the expression of miR159 and *RhCKX6* in the same petals of these

successfully inoculated plants by RT-qPCR and selected the over-expression plants to analyze the phenotypes.

### Statistical analysis

The statistical significance of the data was tested in GraphPad Prism version 8.0 (GraphPad Software Inc., San Diego, CA, USA: http://www.graphpad.com/). All results were tested with Student's *t*-test or one-way analysis of variance (ANOVA) with Tukey's multiple comparisons test.

### Reporting summary

Further information on research design is available in the Nature Portfolio Reporting Summary linked to this article.

## Data availability

Data supporting the findings of this study are available from the corresponding author upon request. The data underlying Figs. 1b–c, e–g; 2e–f; 3c; 4b–f; 5a–g; 6a–b, d–f; 7d–f; 8b–h; 9d–f and Supplementary Figs. 1a–c; 2a; 7c–d; 8a, d; 9b; 10b–d; 12e; 13f–h are provided in the Source Data file. All primers used in this study are listed in Supplementary Data 6. Information on the genes used in this study is given in Supplementary Data 7. RNA-seq data that support the findings of this study have been deposited in the NCBI Bioproject database under accession number PRJNA808873. The raw mass spectrometry data of validation have been deposited to the ProteomeXchange Consortium under accession number PXD046197. Source data are provided with this paper.

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

## Acknowledgements

This work was supported by the National Key Research and Development Program (Grant 2020YFD1000405, J.G., N.M., X.Z.), the National Natural Science Foundation of China (Grant number 31872148 N.M. and 32202530 W.J.), the Construction of Beijing Science and Technology Innovation and Service Capacity in Top Subjects (CEFF-PXM2019_014207_000032, J.G., N.M., X.Z.), and the 111 Project (B17043, J.G., N.M., X.Z.). We thank Dr. Zhizhong Gong (China Agricultural University) for providing the pSuper1300 vector and Dr. Wangjin Lu (South China Agricultural University) for providing the pBD, pBD-VP16, and dual-reporter vectors. We thank Dr. Rui Xia (South China Agricultural University) for offering technical support for the miRNA analysis.

## Author contributions

N.M. and X.Z. designed the research. W.J., F.G., G.L., J.L., Y.D. and W.Y. performed the experiments. X.S., Y.L. and J.G. provided technical support and conceptual advice. W.J., X.Z. and N.M. conducted data analysis. W.J., F.G., X.Z. and N.M. wrote the manuscript.

## Competing interests

The authors declare no competing interests.
