## [Peer Review File · Nature Communications]

Petal size is controlled by the MYB73/TPL/HDA19-miR159-CKX6 module regulating cytokinin catabolism in *Rosa hybrida*REVIEWER COMMENTS

Reviewer #1 (Remarks to the Author):

The authors are exploring the role of miR159 in regulating cytokinin levels to control petal development in rose, using virus-induced gene silencing to deduce roles for miR159, for the cytokinin oxidase CKX6, and for the transcription factor MYB73. The authors propose that these players control the transition from cell division to cell expansion in the petal, however the rose petal system they are using does not actually allow one to draw conclusions about such a transition. They may, however, with a more careful analysis be able to draw some conclusions about the role of these players in how cytokinin regulates cell division.

Major concerns:

1. The authors state that in “petals, cell division mostly ceases before anthesis, and flower opening is largely attributed to cell expansion,” using this transition from bud closed to opening as their definition as to when cell division stops and cell expansion begins.

However, the primary reference (Yamada et al 2009) the authors use for rose petal development contradicts this concept of petal transition, explicitly stating that for rose, “cell division does not stop at later stages of flower opening.” Most interestingly, Yamada et al (Fig. 3) found sharp differences between what occurred for the adaxial and abaxial epidermal cells during petal development. The adaxial cells continued to divide through stage 6 at a fairly consistent rate, such that cell division was a major contributor to the adaxial petal area. In contrast the abaxial cells had largely ceased division by stage 2, well before bud opening, with cell expansion then being a major contributor to the abaxial petal area. Based on this background, the authors cannot infer how cytokinin is regulating the transition from cell division to cell expansion in rose petals.

2. Although the authors cannot use their study to draw conclusions about the role of cytokinin in regulating the transition from cell division to cell expansion, there is the possibility of exploring the role of cytokinin in regulating cell division itself. For this the authors would need to perform a more careful analysis of the number of cells found in the adaxial and abaxial epidermal cell layers. Specifically, the authors only provide analysis at a single time point (e.g. day five) for their analysis of epidermal cell size in the petals, the size of the cells being suggestive as to whether changes in cytokinin activity is affecting cell division. But the authors do not indicate whether the images are from the adaxial or the abaxial surface. Given the sharp differences in the roles of cell division vs cell expansion for these two surfaces (see above), this is critical information and it would be preferable to provide cell size information for both adaxial and abaxial epidermal cells. In addition, due to the changes in timing of development for the control vs gene silenced petals, one would need to compare the petals at the same developmental stage (i.e. stage 6 at which the petal has fully expanded). From this analysis, the authors could measure cell size and petal area, and then determine the total number of cells per petal for the adaxial and abaxial epidermal layers, thereby indicating how much cytokinin is regulating cell division in the petal. Note that cell size alone is not sufficient to elucidate effects on cell division. Problematically, for their figures, the authors give petal size for fully expanded petals (e.g. days 6 and 7) but cell size at day 5 (before the petals are fully expanded), so the authors cannot directly determine the cell number per petal from the data presented in the manuscript.

3. miR159 has a well-characterized role in the targeting of Mybs that regulate the response to GA, the best known such role being to suppress their activity so as to facilitate cell

division. One would thus expect that such Mybs might be under control of miR159 in the rose petal for the same purpose. To this end the authors need to examine more than just CKX6 as a potential target to but also provide expression and cleavage information for the GA Mybs.

4. The GA-mybs are the conserved targets for miR159 across multiple plant species, so the finding that CKX6 is a target in rose represents a non-conserved target. This raises the question as to how common the miR159-CKX6 pathway is as a means of regulating cytokinin levels. If a decrease in cytokinin levels is a common theme during leaf and petal development in different plant species, then it is likely that the expression of CKX's is regulated at the level of transcription not just through miR159 as the authors propose for rose. The authors thus need to clearly distinguish the effects of miR159 from transcriptional regulation during development of the petal. The authors show relative expression of CKX6 in control vs miR159 silenced lines at a single time point (Fig 5e, 3 days after stage 0). But it is unclear how much CKX6 expression changes over time and to what degree this is affected by silencing miR159. For example, the authors show a heatmap from a transcriptional database (Fig 2d) that expression of CKX6 increases about 80-fold from stage 0 to stage 2, but the effects of silencing miR159 only result in a 3-fold increase in CKX6 expression. The authors need to follow CKX6 expression throughout their developmental timecourse for the control and miR159 silenced lines to determine how much of an effect miR159 has on the regulation of CKX6 expression.

Reviewer #2 (Remarks to the Author):

Jing et al report on the regulation of petal size in roses. They find that miR159 regulates the expression of CKX6, which codes for an enzyme degrading the hormone cytokinin. During organ formation, cytokinin regulates the transition of cells from the division phase to differentiation and growth. It is shown that altering the expression of miR159 or CKX6 alters flowering size in predicted ways, corroborating the regulatory connections between the genes and the concept that the miR159/cytokinin module regulates rose flower size. Furthermore, upstream transcriptional and epigenetic regulators of MIR159 expression are detected and their functional relevance is demonstrated. Overall, this is an interesting and new story adding significant new knowledge to the field, the experimental work has been done mostly properly and the article is generally well written. I have some points to be considered.

Title

I think the adjective "Rapid" is not justified as a rapid clearance of cytokinin is not truly shown. The notion "rapid" appears also in abstract and in the main text, it should be considered carefully (the drop in miR159 takes 5 days, line 85, this is not rapid). May be the impact of rose flower size may be mentioned in the title, e.g. add at the end "... and regulates rose flower size"

Abstract

For the second sentence I would argue that the relevance of cytokinin to regulate the transition from cell proliferation to expansion is an established concept, it is well documented in roots and in the shoot meristem, it is not a largely unknown question. Key papers are Werner et al. (Plant Cell 15, 2532-2550 2003) and Bartrina et al. (Plant Cell 23, 69-80. 2011), more details on the root meristem can be found for example in work of the Sabatini

group (Dello Iorio et al., Science 322, 1380-1384, 2008). This background should be considered.

Introduction

The introduction section is rather short. A few sentences about cytokinin (metabolism, maybe signaling, regulation of cell cycle and cell proliferation) should be included, as well as some known functions of miR159 (this can be partly shifted from Discussion section).

Line 37: I doubt that the size of flowers is constant in a given species. There are large differences of flower size between different varieties of species. Please adapt the sentence.

Results

Line 110-115. The claimed transcript analysis of CKX6 is not shown on Fig. 2 but it would be necessary to include a detailed analysis of the changes occurring during petal development.

Line 131: What are the canonical miR159 targets? And what makes up a “canonical” target? Some of that can be found in the Discussion but that information should be given in the Introduction.

Lines 145-146: citation style changed. I could not find any of the here mentioned publications in the reference section.

Lines 188-189: Fig. 7e only shows that RhMYB73 is a transcriptional repressor in general, MIR159 is not included here, right?

Figures

Figs. 1a and b: What has been measured here – the precursor transcript or mature miR159? The y-axis is titled “relative miR159 abundance”, but the legend says “pre-MIR159 levels”. In the Methods section, there is no mentioning of mature miRNA detection.

Fig. 1e: In the Figure legend, only the P values for *** and **** are mentioned, but not for *, which is indicated in this part figure. What exactly is the “flower development progress (d)” shown in Fig. 1c?

Fig. 2. From where are these data? No source is given in the legend. The RhCKX6 transcript analysis should be more extensive, including a kinetic covering the critical developmental period (currently a GFP fusion gene analysis is shown in Fig. 2).

Fig. 3e. Please add a quantification of the signals relative to the controls. In the current version the degree of changes cannot be evaluated.

Fig. 4b Is there information on the degree of changes in relative expression in different phases of flower development?

Fig. 5 There are large differences in cytokinin content in different measurements. Compare Fig. 5b and 5g, there is a 20-fold difference in the tZ content in TRV. How is that explained?

Figs. 7a/S8: Please give a more detailed description of what can be seen here. What do “-, +” and “1x, 10x, 100x” mean exactly?

Discussion

The discussion should be more focused, there are sections reporting rather broadly on general knowledge that can be shortened, part of the background should appear in the introduction.

Line 253. The first heading does not read well, please rephrase.

Line 257/258ff. The transition from division to differentiation involves “multiple phytohormones”. I wonder why here a one-page long summary of other plant hormones is given, which are not considered in this work, while the known facts about cytokinin (see above) are not described. That section should be adapted.

Line 289, “energetically costly”, please explain; the continuation of the sentence is questionably, please think it over.

Lines 307-309: It might be worth mentioning that also cytokinin is involved in juvenile phase control (Werner et al., Nature Comm. 12, 5816, 2021) described here to be linked to a miR159-MYB33 module affecting miR156.

How about the CKX6 genes in other species? Are they predicted targets of miR159? In Arabidopsis?

Please consider the recent paper of Zou et al. on rose CKX6, putting it in the context of petal dehydration and show its regulation by RaNAP transcription factor (Molecular Horticulture 1, 13, 2021)

Methods

Line 394: “Quantification” instead of “quantitative”.

Line 400. 100 mM 6-BA is a very high, presumably (almost) toxic concentration (usually 1-5 μ M are used). Please check if that is correct.

Cytokinin measurements. From how many plants were sthe samples, how many flowers (in approximation)? It would be good to include a table reporting the results for all cytokinin metabolites measured in the supplements, here the data are limited to few metabolites.

Information on how pictures (BiFC, LUC) were evaluated is completely missing.

There are some inconsistencies in units, as both μ g and microgram, mg/L, ng/ml and mg x l⁻¹ are used, please unify.

Reviewer #3 (Remarks to the Author):

Review of Rapid clearance of cytokinin controls transition from cell division to expansion during petal development in rose (*Rosa hybrida*)

MiR159 is an evolutionarily conserved microRNA that regulates GAMYB transcription factors in angiosperms. Here, the authors report that rose miR159 regulates Cytokinin

Oxidase/Dehydrogenase 6 (CKX6) to control petal size. The authors use a target mimicry strategy (TRV-STTM159), to sequester and/or degrade miR159, which resulted in flowers with smaller petals that have fewer cells. Among the predicted targets of miR159 that have increased expression levels after inactivating miR159 by STTM159, the authors found CKX6. In turn, using VIGS to silence CKX6 resulted in changes in cytokinin catabolism and larger petals. Furthermore, the authors show that rhMYB73 binds to the MIR159 promoter repressing its expression during flower development.

In principle, I think that the acquisition of a new target by the evolutionarily conserved miR159 to control petal size in roses is interesting. However, I have concerns about the data supporting the regulation of CKX6 by miR159 and the proposed role of a miR159-CKX6 module.

Main concerns:

1- The authors mention that the underlying mechanism controlling the transition from cell proliferation to expansion in leaves remains largely unknown. However, there are known regulators controlling this transition, such as the miR319-regulated TCP (CIN-Like) transcription factors.

miR319 is similar in sequence to miR159, and target mimicry against miR159 can cross target miR319 (Reichel and Millar, 2015). In addition, miR319-TCP controls petal growth and development (i.e., Nag et al., 2009). Therefore, I think that the authors need to consider the possibility that STTM159 could be affecting petal growth through the modification of the miR319-TCP module.

The complete miR319/miR159 family of miRNAs in rose (which might contain several members) should be analyzed in STTM159 plants. The analysis of miRNA-targets should also include rose TCP transcription factors that are targets of miR319.

2- MicroRNAs guide the cleavage of their targets at position 10-11 (Fig 3a). The authors did not find any cut at this position for CKX6, which argues against the regulation by miR159. The authors say that they analyzed CKX6 cleavage in rose petals in the Results section, but in Methods they say that the experiment was performed from 35S:pre-MIR159 transient overexpression plants (tobacco?). The cleavage site should be determined in wt rose petals.

miR159 is strongly expressed in tobacco leaves, so I'm surprised that the authors could express large amounts of a GFP sensor for a miR159-target (Fig 3). The levels of mature miR159 should be determined in tobacco leaves to show the increase of miR159 after the expression of rose pre-miR159.

The determination of mature miRNAs by qPCR can be tricky – for example, miRNAs from the same family can have differences in their 3' end, yet they regulate the same targets. It is better to determine miRNA levels by small RNA blots.

The expression of the sensors in *N. benthamiana* leaves is patchy which might be caused by RNA silencing (independent of the miRNA).

Overall, I don't think that the current manuscript shows clear evidence for the regulation of CKX6 by miR159.

3- Inactivation of miR159 (by target mimicry or mir159a/b knock outs) in Arabidopsis causes pleiotropic developmental defects in leaves and stunted plants. Please, show and describe the plant growth and overall phenotypes of the VIGS lines (STTMIR159, and rhMYB73 and RhCKX6)-silenced lines. Also, indicate the number of plants analyzed in each case, and the distribution of the phenotypes observed (i.e., percentage of plants displaying the selected phenotypes).

4- For VIGS, authors used fragments of approx. 400 bp for RhMYB73M159 and RhCKX6. These are long fragments. Are the authors confident that other MYBs or CKX genes were not affected? I think using artificial microRNAs would have been a more specific strategy.

I think it would be important to include the phenotypes of roses overexpressing pre-miR159, which should down-regulate RhCKX6 and increase petal size. This vector is used in the *N. benthamiana* assays.

5- The authors predict the regulation of several CKXs by miR159 in rose and arabidopsis (Supplementary Fig 4). The location of a miRNA-target site in a target gene is usually conserved during evolution and not variable as depicted in this figure. There is plenty of experimental information identifying miRNA targets in arabidopsis in publicly available databases. Please use this info to show the empirical data for the regulation by miR159 for the two predicted Arabidopsis targets.

Other comments:

1- Please include in the Supplementary Tables the description for the selected rose genes (or the description of their arabidopsis homologues).

2- L78 –“To understand the underlying mechanism of the transition between cell division and cell expansion, we previously conducted a transcriptome deep sequencing (RNA-seq) analysis of short RNAs and transcripts from petals to investigate gene expression profiles during the cell division and expansion phases.” – please include the citation for the article describing the RNAseq analysis of short RNAs and transcripts.

Response to Reviewers

Manuscript No. NCOMMS-22-37721

Title: Rapid clearance of cytokinin controls transition from cell division to expansion during petal development in rose (*Rosa hybrida*)

Reviewer #1

Comment #1

The authors state that in “petals, cell division mostly ceases before anthesis, and flower opening is largely attributed to cell expansion,” using this transition from bud closed to opening as their definition as to when cell division stops and cell expansion begins. However, the primary reference (Yamada et al 2009) the authors use for rose petal development contradicts this concept of petal transition, explicitly stating that for rose, “cell division does not stop at later stages of flower opening.” Most interestingly, Yamada et al (Fig. 3) found sharp differences between what occurred for the adaxial and abaxial epidermal cells during petal development. The adaxial cells continued to divide through stage 6 at a fairly consistent rate, such that cell division was a major contributor to the adaxial petal area. In contrast the abaxial cells had largely ceased division by stage 2, well before bud opening, with cell expansion then being a major contributor to the abaxial petal area. Based on this background, the authors cannot infer how cytokinin is regulating the transition from cell division to cell expansion in rose petals.

Although the authors cannot use their study to draw conclusions about the role of cytokinin in regulating the transition from cell division to cell expansion, there is the possibility of exploring the role of cytokinin in regulating cell division itself. For this the authors would need to perform a more careful analysis of the number of cells found in the adaxial and abaxial epidermal cell layers. Specifically, the authors only provide analysis at a single time point (e.g. day five) for their analysis of epidermal cell size in the petals, the size of the cells being suggestive as to whether changes in cytokinin activity is affecting cell division. But the authors do not indicate whether the images are from the adaxial or the abaxial surface. Given the sharp differences in the roles of cell division vs

cell expansion for these two surfaces (see above), this is critical information and it would be preferable to provide cell size information for both adaxial and abaxial epidermal cells.

In addition, due to the changes in timing of development for the control vs gene silenced petals, one would need to compare the petals at the same developmental stage (i.e. stage 6 at which the petal has fully expanded). From this analysis, the authors could measure cell size and petal area, and then determine the total number of cells per petal for the adaxial and abaxial epidermal layers, thereby indicating how much cytokinin is regulating cell division in the petal. Note that cell size alone is not sufficient to elucidate effects on cell division. Problematically, for their figures, the authors give petal size for fully expanded petals (e.g. days 6 and 7) but cell size at day 5 (before the petals are fully expanded), so the authors cannot directly determine the cell number per petal from the data presented in the manuscript.

RESPONSE

Thank you for your excellent advice.

We would like to apologize for any confusion caused by the incomplete and misleading description in our previous version. We would like to clarify that all data presented was based on observations of abaxial epidermal cells. Our previous study utilized scanning electron microscopy to observe the anatomical structure of rose petals. We found that cell number and size of abaxial epidermal cells are closely related to petal size, which is why we focused our analysis on this particular cell type (Ma et al., 2008, Plant Physiology).

In our previous version, we tried to demonstrate that abaxial epidermal cells in TRV-*STTM159* lines entered the expansion stage earlier compared to TRV controls. To this end, we illustrated the cell size on day 5 (Fig.1f) and on the fully opened stage (Supplementary Fig.2b).

According to your advice, we conducted further experiments, wherein we recorded the number of adaxial and abaxial epidermal cells in TRV controls and different gene silenced lines (TRV-*STTM159*, TRV-*RhCKX6*, and TRV-*RhMYB73*), following the procedure outlined by Yamada et al. (2009). Our findings suggested that silencing of miR159 resulted in a slower increase in the number of adaxial and abaxial epidermal cells, particularly in abaxial epidermal cells. Furthermore, the cell division period was reduced in TRV-*STTM159* lines, as the plants in these lines attained full opening earlier than TRV controls. In contrast, silencing of either *RhCKX6* or *RhMYB73* showed an opposite

phenotype to TRV-*STTM159* lines.

We added these data in the revised manuscript as Supplemental figure S1; Fig. 1f, g; Fig. 4e, f; and Fig. 8e,f. We also revised the title and our manuscript accordingly to avoid any misleading in the revised version.

Figure 1 miR159 controls cell division and flower opening in rose.

f, g. The cell number of adaxial epidermis (f) and abaxial epidermis (g) in TRV control and TRV-*STTM159* petals from days 1 after floral stage 0 to fully opened stage. Data are shown as means \pm SD from 3 biological replicates (n = 3).

Please refer to the revised version for other Figures.

Comment #2

miR159 has a well-characterized role in the targeting of Mybs that regulate the response to GA, the best known such role being to suppress their activity so as to facilitate cell division. One would thus expect that such Mybs might be under control of miR159 in the rose petal for the same purpose. To this end the authors need to examine more than just CKX6 as a potential target to but also provide expression and cleavage information for the GA Mybs.

RESPONSE

Thank you for your advice. According to your suggestion, we isolated the homologues of all seven Arabidopsis GAMYB genes (*MYB33*, *MYB65*, *MYB81*, *MYB97*, *MYB101*, *MYB104*, and *MYB120*). We predicted miR159 targets in three of them, including *RhMYB33*, *RhMYB65*, and *RhMYB101*. Both RNA-seq and qRT-PCR analysis showed that expression of *RhMYB65* and *RhMYB101* could not be detected in rose petals in all tested development stages (1 d, 3 d, and 5 d). Expression of *RhMYB33* was detectable but was much lower than expression level of *RhCKX6*. In addition,

expression of *RhMYB33* slightly decreased in petals from 3 d to 7 d. RLM-RACE showed the miR159-dependent cleavage of *RhMYB33* transcript in rose petals. We constructed an *RhMYB33*-Sensor and an *RhMYB33m*-Sensor constructs by cloning the coding sequence of green fluorescent protein (GFP) in-frame with an intact miR159-target region (for *RhMYB33*-Sensor) or a mutated miR159-target region (for *RhMYB33m*-Sensor). We then co-infiltrated each sensor construct in *Nicotiana benthamiana* leaves with a construct overexpressing the miR159 precursor (pre-*MIR159*) and detected lower GFP fluorescence from the pre-*MIR159*+*RhMYB33*-Sensor combination compared to the Sensor alone, but not from the pre-*MIR159*+*RhMYB33m*-Sensor combination. In addition, we tested GAs level in rose petals. Level of GA3 and GA7 was barely detectable, while level of GA1 and GA19 remained stable throughout the developmental time course. Based on these data, we considered that *GAMYBs*-mediated regulation of cell division is insignificant relative to *RhCKX6*-mediated regulation, although we could not completely exclude the possibility that miR159-*RhMYB33* module is involved in regulation of petal cell division in roses. We added these data as Supplemental Figure S7 and discussed this issue in the revised version.

Supplemental Figure S7 miR159-targeted *RhMYB33* and GAs contents in rose petals during petal growth.

a, Upper panel, schematic diagram of *RhMYB33* mRNA. Red box, predicted cleavage site of miR159. Bottom panel,

identification of cleavage sites using 5' RLM-RACE assay in rose petals. The positions of cleavage sites are indicated by arrowheads with the frequency of clones.

b, Confocal imaging analysis of *N. benthamiana* leaves 3 days after co-infiltration of *pre-MIR159* with *RhMYB33-sensor-GFP* or *RhMYB33m-sensor-GFP*, respectively. GFP fluorescence of *N. benthamiana* leaves 3 days after co-infiltration of the indicated constructs. The experiment was performed independently three times, and representative results are shown. Scale bars, 5 mm.

c, RT-qPCR analysis of *RhMYB33* transcript levels in petals 1, 3, 5 days after stage 0. *RhUBI2* was used as an internal control. Data are shown as means \pm SD from three biological replicates ($n = 3$). Different lowercase letters above each bar indicate significant differences according to one-way ANOVA with Tukey's multiple comparisons test ($P < 0.05$).

d, GA contents in petals 1, 3, 5 days after stage 0. Data are shown as means \pm SD from three biological replicates ($n = 3$).

Comment #3

The GA-mybs are the conserved targets for miR159 across multiple plant species, so the finding that CKX6 is a target in rose represents a non-conserved target. This raises the question as to how common the miR159-CKX6 pathway is as a means of regulating cytokinin levels. If a decrease in cytokinin levels is a common theme during leaf and petal development in different plant species, then it is likely that the expression of CKX's is regulated at the level of transcription not just through miR159 as the authors propose for rose. The authors thus need to clearly distinguish the effects of miR159 from transcriptional regulation during development of the petal. The authors show relative expression of CKX6 in control vs miR159 silenced lines at a single time point (Fig 5e, 3 days after stage 0). But it is unclear how much CKX6 expression changes over time and to what degree this is affected by silencing miR159. For example, the authors show a heatmap from a transcriptional database (Fig 2d) that expression of CKX6 increases about 80-fold from stage 0 to stage 2, but the effects of silencing miR159 only result in a 3-fold increase in CKX6 expression. The authors need to follow CKX6 expression throughout their developmental time course for the control and miR159 silenced lines to determine how much of an effect miR159 has on the regulation of CKX6 expression.

RESPONSE

Thank you for your excellent suggestion. We detected expression level of miR159 and *RhCKX6* in petals throughout the developmental time course for the TRV control and TRV-*STTM159* lines. The results showed that expression of *RhCKX6* in TRV-*STTM159* lines was significantly higher than TRV control throughout the developmental time course, especially in the earlier period. Meanwhile,

expression of *RhCKX6* increased in TRV-*STTM159* lines in the development duration.

We considered this might be two reasons, 1) the VIGS just knockdown the miR159 instead of knockout. The rest miR159 still functioned to influence accumulation of *RhCKX6* transcripts. 2) Besides of miR159-mediated regulation, expression of *RhCKX6* might be induced by some unknown factors during petal development as well.

In addition, we tested the possible miR159-mediated cleavage of *CKXs* in more plants. We predicted miR159-target site in *AtCKX5* (*Arabidopsis*), *MdCKX6* (*Malus domestica*), and *FvCKX6* (*Fragaria vesca*). We generated GFP-Sensor constructs for *AtCKX5* (*Arabidopsis*), *MdCKX6* (*Malus domestica*), and *FvCKX6*, and detected miR159-mediated cleavage of these three Sensors. RLM-RACE also confirmed miR159-mediated cleavage of *FvCKX6* in petals of *Fragaria vesca*, indicating that miR159-mediated cleavage of *CKXs* could be a common regulation pathway.

Based on these data, we considered miR159 played an essential role in regulation of accumulation of *RhCKX6* transcripts, but we could not exclude the possibility that some other currently unknown factors might regulate expression of *RhCKX6* as well.

We added the results as Fig. 2e-g and Supplemental Figure S6, toned down our statement and discussed this issue in the revised version.

Figure 2 miR159 influences transcript accumulation of cytokinin catabolism genes.

e, f, RT-qPCR analysis of miR159 abundance and *RhCKX6* transcript levels in rose petals at 1, 3, 5, and 7 days after stage 0. *5S rRNA* was used as an internal control (**e**). *RhUBI2* was used as an internal control (**f**). Data are shown as means \pm SD are shown from three biological replicates ($n = 3$). Different lowercase letters above each bar in (**e** and **f**) indicate significant differences according to one-way ANOVA with Tukey's multiple comparisons test ($P < 0.05$). **g**, RT-qPCR analysis of *RhCKX6* transcript levels in petals of TRV control and TRV-*STTM159* at 1, 3, and 5 days after stage 0. *RhUBI2* was used as an internal control. Data are shown as means \pm SD are shown from three biological replicates ($n = 3$). Asterisks indicate statistically significant differences (two-sided Student's *t*-test, *, $P < 0.05$; **, $P < 0.01$; ns, no significant difference).

Reviewer #2

Comment #1

Title

I think the adjective “Rapid” is not justified as a rapid clearance of cytokinin is not truly shown. The notion “rapid” appears also in abstract and in the main text, it should be considered carefully (the drop in miR159 takes 5 days, line 85, this is not rapid). May be the impact of rose flower size may be mentioned in the title, e.g. add at the end “... and regulates rose flower size”

RESPONSE

Many thanks for your excellent advice. As your suggestion, we have revised the title to “In rose, miR159 regulates petal cell division by modulating cytokinin catabolism”.

Comment #2

Abstract

For the second sentence I would argue that the relevance of cytokinin to regulate the transition from cell proliferation to expansion is an established concept, it is well documented in roots and in the shoot meristem, it is not a largely unknown question. Key papers are Werner et al. (Plant Cell 15, 2532-2550 2003) and Bartrina et al. (Plant Cell 23, 69-80. 2011), more details on the root meristem can be found for example in work of the Sabatini group (Dello Ioio et al., Science 322, 1380-1384, 2008). This background should be considered.

RESPONSE

Many thanks for your excellent comments, which makes our description more exact. In abstract, we have edited the description. In addition, we have revised our Introduction section by adding the information of the key papers you suggested.

Comment #3

Introduction

The introduction section is rather short. A few sentences about cytokinin (metabolism, maybe signaling, regulation of cell cycle and cell proliferation) should be included, as well as some known

functions of miR159 (this can be partly shifted from Discussion section).

Line 37: I doubt that the size of flowers is constant in a given species. There are large differences of flower size between different varieties of species. Please adapt the sentence.

RESPONSE

Many thanks for your advice. We have revised the Introduction section by adding the information of cytokinin and function of miR159.

We have deleted the sentence “*In plants, the size of flowers and petals is constant within a certain species, but varies across different species*” in the revised version.

Comment #4

Results

Line 110-115. The claimed transcript analysis of CKX6 is not shown on Fig. 2 but it would be necessary to include a detailed analysis of the changes occurring during petal development.

RESPONSE

Thank you for your advice. We would like to apologize for the missing data. We have added the expression level of *RhCKX6* in Fig. 2f in the revised version.

Comment #5

Line 131: What are the canonical miR159 targets? And what makes up a “canonical” target? Some of that can be found in the Discussion but that information should be given in the Introduction.

RESPONSE

Many thanks for your suggestions. As you suggested, we have added the relevant description of the canonical miR159 targets in the Introduction section in the revised version.

Comment #6

Lines 145-146: citation style changed. I could not find any of the here mentioned publications in the reference section.

RESPONSE

Thank you for your advice. We would like to apologize for this mistake. We have corrected the citation and the Reference list.

Comment #7

Lines 188-189: Fig. 7e only shows that RhMYB73 is a transcriptional repressor in general, MIR159 is not included here, right?

RESPONSE

Thank you for pointing this out. As you mentioned, Fig. 7e only shows that RhMYB73 is a transcriptional repressor in general, MIR159 is not included here. We have corrected the description in the revised version.

Comment #8

Figures

Figs. 1a and b: What has been measured here – the precursor transcript or mature miR159? The y-axis is titled “relative miR159 abundance”, but the legend says “pre-MIR159 levels”. In the Methods section, there is no mentioning of mature miRNA detection.

Fig. 1c: In the Figure legend, only the P values for *** and **** are mentioned, but not for *, which is indicated in this part figure. What exactly is the “flower development progress (d)” shown in Fig. 1c?

RESPONSE

Thank you for your guidance. We would like to apologize for this mistake. Actually, both Figure 1a and b presented the mature miR159 abundance. We replaced the Figure 1a as the Northern Blot detection of mature miR159 in the revised version and moved the original Figure 1a to Supplementary Figure 2a. We added the methods of Northern Blot and qRT-PCR of mature miR159 in the Methods section in the revised version. We also revised the Figure legend and mentioned “*”. The “flower development progress (d)” is the period of different stage of flower opening. We have updated the y-axis label of Figure 1c from "flower development progress (d)" to "Period of different

stage of flower opening (d)" to more accurately represent the data. The y-axis label of Figure 4c, 6b, 8b, and Supplemental Fig. S10b were revised accordingly.

Comment #9

Fig. 2. From where are these data? No source is given in the legend. The *RhCKX6* transcript analysis should be more extensive, including a kinetic covering the critical developmental period (currently a GFP fusion gene analysis is shown in Fig. 2).

RESPONSE

Thank you for your guidance. We have included the source information in the Legend of Figure 2 and added the expression of the *RhCKX6* gene in petals during petal development in Fig. 2f in the revised version.

Comment #10

Fig. 3e. Please add a quantification of the signals relative to the controls. In the current version the degree of changes cannot be evaluated.

RESPONSE

Thank you for your guidance. We have added the quantification of the signals relative to the controls in Fig. 3d in the revised version.

Comment #11

Fig. 4b Is there information on the degree of changes in relative expression in different phases of flower development?

RESPONSE

Thank you for your guidance. Fig. 4b is the expression of *CKX6* in *CKX6*-silencing plants (TRV-*RhCKX6*) and control (TRV), which is test the degree of *CKX6*-silencing. We have added the expression of the *RhCKX6* gene in petals during petal development in Fig. 2f in the revised version.

Comment #12

Fig. 5 There are large differences in cytokinin content in different measurements. Compare Fig. 5b and 5g, there is a 20-fold difference in the tZ content in TRV. How is that explained?

RESPONSE

Thank you for pointing this out. The difference in tZ content observed in the TRV shown in Fig. 5b and 5g was due to the difference in the sampling time. In Fig. 5b, the petals were sampled for the determination of tZ content 5 days after stage 0, whereas in Fig. 5g, the petals were sampled 3 days after stage 0. According to Fig. 6a, the tZ content in the petals significantly reduces on the 5th day after stage 0, as compared to the 1st day. It is noteworthy that the petals were sampled on the 3rd day after stage 0 in Fig. 5g because the silencing of miR159 caused a shortened cell division period. We have revised the relevant description to avoid any misleading.

Comment #13

Figs. 7a/S8: Please give a more detailed description of what can be seen here. What do “-, +” and “1x, 10x, 100x” mean exactly?

RESPONSE

Thank you for your guidance. We have added the description of “-, +” and “1x, 10x, 100x” in Figs. 7a/S11 legends.

Comment #14

Discussion

The discussion should be more focused, there are sections reporting rather broadly on general knowledge that can be shortened, part of the background should appear in the introduction.

Line 253. The first heading does not read well, please rephrase.

RESPONSE

Many thanks for your excellent advice. We have revised the Discussion section according to your guidance. We deleted some general information and moved part of the background to the Introduction section. We have revised the first heading to “*Cytokinin accumulation is vital to lateral*

organ development through controlling the duration of cell division”.

Comment #15

Line 257/258ff. The transition from division to differentiation involves “multiple phytohormones”. I wonder why here a one-page long summary of other plant hormones is given, which are not considered in this work, while the known facts about cytokinin (see above) are not described. That section should be adapted.

RESPONSE

Many thanks for your advice. We have deleted the un-relevant description and made it more focused on cytokinin.

Comment #16

Line 289, “energetically costly”, please explain; the continuation of the sentence is questionably, please think it over.

RESPONSE

Many thanks for your advice. The phrase "energetically cost" is used to describe the negative regulation of cytokinin levels by miR159-mediated cleavage of *RhCKX6*. This regulatory pattern involves *RhCKX6* transcripts being continuously produced, followed by their subsequent cleavage by miR159 in the early stages of petal development. As a result, considerable energy is expended to generate these transcripts, making the process energetically costly. We deleted this sentence to avoid any misleading in the revised version.

Comment #17

Lines 307-309: It might be worth mentioning that also cytokinin is involved in juvenile phase control (Werner et al., Nature Comm. 12, 5816, 2021) described here to be linked to a miR159-MYB33 module affecting miR156.

RESPONSE

Many thanks for your advice. We have revised our Discussion section by adding the information of

the report of Werner et al. (2021).

Comment #18

How about the CKX6 genes in other species? Are they predicted targets of miR159? In Arabidopsis?

RESPONSE

Thank you for your guidance.

We conducted further experiments to examine the possibility of miR159-mediated cleavage of *CKXs* in other plant species. Using bioinformatics analysis, we identified a miR159-target site in *AtCKX5* (Arabidopsis), *MdCKX6* (*Malus domestica*), and *FvCKX6* (*Fragaria vesca*). We then created GFP-Sensor constructs for each of these CKXs, following the procedure described in our response to Comment #3 of reviewer 1. Our results revealed that miR159-mediated cleavage occurred for all three Sensors. To confirm this result, we performed RLM-RACE and found evidence of cleavage in *FvCKX6* in the petals of *Fragaria vesca*. These findings suggest that miR159-mediated cleavage of *CKXs* represents a common regulation pathway across different plant species.

We have added these results as Supplemental Figure S6, and mentioned the results in the revised version.

Comment #19

Please consider the recent paper of Zou et al. on rose CKX6, putting it in the context of petal dehydration and show its regulation by RaNAP transcription factor (Molecular Horticulture 1, 13, 2021)

RESPONSE

Many thanks for your advice. We have added the relevant information of the report of Zou et al. (2021) in the Discussion section.

Comment #20

Methods

Line 394: “Quantification” instead of “Quantitative”.

RESPONSE

Many thanks for your advice. We have replaced “Quantitative” with “Quantification”.

Comment #21

Line 400. 100 mM 6-BA is a very high, presumably (almost) toxic concentration (usually 1-5 μ M are used). Please check if that is correct.

RESPONSE

Thank you for pointing this out. We would like to apologize for this mistake. It should be 100 μ M here. The concentration of 6-BA used and treatment procedure were followed a previous report (Wu et al., 2017). We have corrected the description and added relevant reference.

References

Lin Wu, Nan Ma, Yangchao Jia, Yi Zhang, Ming Feng, Cai-Zhong Jiang, Chao Ma, Junping Gao.

An ethylene-induced regulatory module delays flower senescence by regulating cytokinin content. *Plant Physiology*, 2017, **173**: 853–862.

Comment #22

Cytokinin measurements. From how many plants were the samples, how many flowers (in approximation)? It would be good to include a table reporting the results for all cytokinin metabolites measured in the supplements, here the data are limited to few metabolites.

RESPONSE

Many thanks for your advice. To conduct the cytokinin test, we collected five flowers from five plants, and mixed them together. The mixed flowers were then divided into three separate samples for testing. We have added the sampling information in the Method section. In addition, we have added the results for all cytokinin metabolites measured as Supplementary Table S3 in the revised version.

Comment #23

Information on how pictures (BiFC, LUC) were evaluated is completely missing.

RESPONSE

Many thanks for your advice. These information have been added in the revised Methods of Supplemental files.

Comment #24

There are some inconsistencies in units, as both μg and microgram, mg/L , ng/ml and mg x l-1 are used, please unify.

RESPONSE

Many thanks for your advice. We have unified the units in the revised manuscript.

Reviewer #3

Comment #1

The authors mention that the underlying mechanism controlling the transition from cell proliferation to expansion in leaves remains largely unknown. However, there are known regulators controlling this transition, such as the miR319-regulated TCP (CIN-Like) transcription factors.

miR319 is similar in sequence to miR159, and target mimicry against miR159 can cross target miR319 (Reichel and Millar, 2015). In addition, miR319-TCP controls petal growth and development (i.e., Nag et al., 2009). Therefore, I think that the authors need to consider the possibility that STTM159 could be affecting petal growth through the modification of the miR319-TCP module. The complete miR319/miR159 family of miRNAs in rose (which might contain several members) should be analyzed in STTM159 plants. The analysis of miRNA-targets should also include rose TCP transcription factors that are targets of miR319.

RESPONSE

Thank you for your excellent advice. According to your valuable suggestion, we have revised our statement, and now we have included the relevant reference of miR319-TCP in the revised version of our manuscript. Furthermore, we conducted Northern blot and qRT-PCR experiments to detect the levels of mature miR159 and miR319 in rose petals during the developmental time course. The results of our Northern blot revealed a significant reduction in miR159 levels from day 1 to day 5,

whereas miR319 levels remained relatively constant from day 1 to day 3 and decreased on day 5. Our qRT-PCR data indicated that the expression level of miR319 in rose petals was much lower than that of miR159 from day 1 to day 5. These results were consistent with our previous miRNA-seq analysis of rose petals during earlier developmental stages (Pei et al., 2013).

In Arabidopsis, miR319/TCPs controls floral organ morphology. Expression of the miR319-resistant form of TCP4 under the control of the petal- and stamen-specific AP3 promoter led to a complete absence of petals and stamens (Nag et al., 2009).

After analyzing our RNA-seq data, we have found that the expression levels of *TCP2*, *TCP4*, and *TCP4-x1*, which are targeted by miR319, were not significantly altered in the petals of the miR159-silenced lines. Upon further confirmation using qRT-PCR, we have determined that *TCP2/4/4-x1* expression levels remained unchanged in the petals of the miR159-silenced lines.

We have added relevant results as Figure 1a and Supplementary Figure S8, and discussed the possible role of miR319/TCPs in the Discussion section.

Supplemental Figure S8 Northern Blot of miR319 and expression of *TCP* genes in petals of miR159-silenced lines.

a, RT-qPCR analysis of miR159 and miR319 levels in rose petals (1, 3 and 5 days after stage 0). Data are shown as means \pm SD from four biological replicates ($n=3$). *5S rRNA* was used as an internal control. Asterisks indicate statistically significant differences (two-sided Student's *t*-test, *, $P < 0.05$; **, $P < 0.01$; ***, $P < 0.001$; ns, no significant difference).

b, Northern blotting analysis of miR319 abundance in rose petals 1, 3, and 5 day after floral bud reached stage 0. *U6* was used as control.

c, Heatmap analysis of miR159 and miR319 abundance in rose petals at stage 0 and stage 2.

d, RT-qPCR analysis of *RhTCPs* in TRV-*STTM159*. *RhUBI2* was used as an internal control. Data are shown as means \pm SD from three biological replicates (n = 3).

References

Haixia Pei, Nan Ma, Jiwei Chen, Yi Zheng, Ji Tian, Jing Li, Shuai Zhang, Zhangjun Fei, Junping Gao. Integrative analysis of miRNA and mRNA profiles in response to ethylene in rose petals during flower opening. PLoS One, 2013, **8**: e64290.

Anwesha Nag, Stacey King, & Thomas Jack. miR319a targeting of TCP4 is critical for petal growth and development in Arabidopsis. Proc. Natl Acad. Sci. USA, 2009, **106**: 22534–22539.

Comment #2

1) MicroRNAs guide the cleavage of their targets at position 10-11 (Fig 3a). The authors did not find any cut at this position for CKX6, which argues against the regulation by miR159. The authors say that they analyzed CKX6 cleavage in rose petals in the Results section, but in Methods they say that the experiment was performed from 35S:pre-MIR159 transient overexpression plants (tobacco?). The cleavage site should be determined in wt rose petals.

2) miR159 is strongly expressed in tobacco leaves, so I'm surprised that the authors could express large amounts of a GFP sensor for a miR159-target (Fig 3). The levels of mature miR159 should be determined in tobacco leaves to show the increase of miR159 after the expression of rose pre-miR159. The expression of the sensors in *N. benthamiana* leaves is patchy which might be caused by RNA silencing (independent of the miRNA).

3) The determination of mature miRNAs by qPCR can be tricky – for example, miRNAs from the same family can have differences in their 3' end, yet they regulate the same targets. It is better to determine miRNA levels by small RNA blots.

Overall, I don't think that the current manuscript shows clear evidence for the regulation of CKX6 by miR159.

RESPONSE

Thank you for your excellent advice.

1) We would like to apologize for this mistake. For Figure 3a, the RLM-RACE was performed using rose petals and we corrected it in the revised version. According to your advice, we sequenced more

clones for RLM-RACE of *RhCKX6* in rose petals. In addition, we predicted miR159-target site in *FvCKX6* (*Fragaria vesca*). RLM-RACE supported the miR159-mediated cleavage of *FvCKX6* in petals of *Fragaria vesca*. These results indicated that miR159-mediated cleavage of *CKX6* could be a common regulation pathway.

Supplemental Figure S6 miR159 targets *FvCKX6* in planta.

a, Confocal imaging analysis of *N. benthamiana* leaves 3 days after co-infiltration of *pre-MIR159* with *FvCKX6-sensor-GFP* or *FvCKX6m-sensor-GFP*.

b, Validation of miR159-targeted cleavage of *FvCKX6*. Upper panel, schematic diagram of *FvCKX6* mRNA. Red box, predicted cleavage site of miR159. Bottom panel, identification of cleavage sites using 5' RLM-RACE assay in strawberry petals. The positions of cleavage sites are indicated by arrowheads with the frequency of clones.

2) According to your suggestions, we detected the level of mature miR159 in tobacco leaves overexpressed rose pre-miR159. The data indicates that the abundance of mature miR159 in tobacco leaves expressing pSuper:pre-MIR159 was significantly higher as compared to those expressing pSuper empty vector.

To prevent gene silencing, a P19 silencing suppressor was co-infiltrated in all combinations in the GFP-Sensor test. The intensity of GFP fluorescence was found to reduce in the pSuper:pre-MIR159+RhCKX6-Sensor combination, but not in the pSuper:pre-MIR159+RhCKX6m-Sensor combination (Fig. 3b). This supports that the decrease in intensity of GFP fluorescence was due to cleavage of RhCKX6-Sensor caused by miR159.

The expression of mature miR159 in tobacco leaves overexpressed rose pre-miR159

RT-qPCR analysis of the mature miR159 in tobacco leaves overexpressed rose pre-miR159. Data are shown as means \pm SD from four biological replicates (n=3). 5S rRNA was used as an internal control.

3) As you suggested, the levels of mature miR159 and miR319 were measured in rose petals throughout petal growth using Northern blot analysis. The results revealed a significant reduction in miR159 levels from day 1 to day 5, whereas miR319 levels remained relatively constant from day 1 to day 3 and decreased on day 5.

Based on the data, it can be inferred that miR159 may have a regulatory role in the accumulation of RhCKX6 transcript levels in rose petals.

We have added these relevant results as in Figure 1a, Supplemental Figure S6 and Supplementary Figure S8, and revised relevant description in the revised manuscript.

Comment #3

Inactivation of miR159 (by target mimicry or mir159a/b knock outs) in Arabidopsis causes pleiotropic developmental defects in leaves and stunted plants. Please, show and describe the plant growth and overall phenotypes of the VIGS lines (STTMIR159, and rhMYB73 and RhCKX6)-silenced lines. Also, indicate the number of plants analyzed in each case, and the distribution of the phenotypes observed (i.e., percentage of plants displaying the selected phenotypes).

RESPONSE

Thank you for your advice. For VIGS, the tissue-cultured plantlets were used for *Agrobacterium-*

mediated infection as described before (Sha et al., 2014; Tian *et al.*, 2014; Chen *et al.*, 2021). Since TRV virus just moved upward and induced gene silencing in the newly grown leaves and flowers after infiltration, we could only observe the phenotypes in these newly emerged leaves and flowers. We did not find that there are stunted newly emerged leaves and plants in TRV-*STTM159*, TRV-*RhCKX6* and TRV-*RhMYB73* silencing lines compared with control (TRV line). According to your valuable suggestions, we described the number of plants displaying the selected phenotypes for each VIGS lines in the VIGS method.

References

- Aihua Sha, Jinping Zhao, Kangquan Yin, Yang Tang, Yan Wang, Xiang Wei, Yiguo Hong, Yule Liu, Virus-Based MicroRNA Silencing in Plants, *Plant Physiology*, 2014, **164**: 36–47
- Ji Tian, Haixia Pei, Shuai Zhang, Jiwei Chen, Wen Chen, Ruoyun Yang, Yonglu Meng, Jie You, Junping Gao, Nan Ma*, TRV-GFP: a modified Tobacco rattle virus vector for efficient and visualizable analysis of gene function, *Journal of Experimental Botany*, 2014, **65**: 311–322.
- Jiwei Chen, Yang Li, Yonghong Li, Yuqi Li, Yi Wang, Chuyan Jiang, Patrick Choisy, Tao Xu, Youming Cai, Dong Pei, Cai-Zhong Jiang, Su-Sheng Gan, Junping Gao, Nan Ma. AUXIN RESPONSE FACTOR 18–HISTONE DEACETYLASE 6 module regulates floral organ identity in rose (*Rosa hybrida*). *Plant Physiology*, 2021, **186**: 1074-1087.

Comment #4

For VIGS, authors used fragments of approx. 400 bp for *RhMYB73* miR159 and *RhCKX6*. These are long fragments. Are the authors confident that other MYBs or CKX genes were not affected? I think using artificial microRNAs would have been a more specific strategy.

RESPONSE

Thank you for your excellent advice. For VIGS, the gene fragment insert is usually in the range of 300–500 nucleotides to ensure efficient silencing of endogenous genes (Burch-Smith et al., 2004; Senthil-Kumar & Mysor, 2011). We chose the gene-specific UTR region to construct the TRV-*MYB73* and TRV-*CKX6* to avoid cross-silencing.

According to your suggestions, we tested expression of two *MYBs* (*RhMYB70/77*), which are close to *RhMYB73*, in *RhMYB73*-silenced lines. Similarly, we tested the expression of two *CKXs*

(*RhCKX1/5*), which are close to *RhCKX6*, in *RhCKX6*-silenced lines. We didn't find significant change of expression of *RhMYB70/77* in *RhMYB73*-silenced lines compared to TRV control, indicating that *RhMYB73* was specifically silenced. Similarly, the results indicated that *RhCKX6* was specifically silenced. We added these results as Supplemental Figure S9 and Supplemental Figure S12d, e.

Supplemental Figure S9 Expression of *RhCKX1* and *RhCKX5* in TRV and *RhCKX6*-silenced lines.

a, Schematic representation of the gene-specific fragment of *RhCKX6* for construction of the TRV-*CKX6* vector.

b, Expression of *RhCKX1* and *RhCKX5* in TRV and *RhCKX6*-silenced plants. *RhUBI2* was used as an internal control. Data are shown as means \pm SD from three biological replicates ($n = 3$) (two-sided Student's *t*-test, ns, no significant difference).

Supplemental Figure S12 Characterization of *RhMYB73*.

d, Schematic representation of the gene-specific fragment of *RhMYB73* for construction of the TRV-*MYB73* vector.

e, Expression of *RhMYB70* and *RhMYB77* in *RhMYB73*-silenced plants. *RhUBI2* was used as an internal control. Data are shown as means \pm SD from three biological replicates ($n = 3$) (two-sided Student's *t*-test, ns, no significant difference).

Meanwhile, we agreed with you that artificial microRNAs could be a more specific strategy, but we could not generate stable transgenic artificial miR159 lines in a short time.

References

Tessa M. Burch-Smith, Jeffrey C. Anderson, Gregory B. Martin, S. P. Dinesh-Kumar. Applications and advantages of virus-induced gene silencing for gene function studies in plants. *The Plant Journal*, 2004, **39**: 734-746.

Muthappa Senthil-Kumar, Kirankumar S. Mysor. New dimensions for VIGS in plant functional genomics. *Trends in Plant Science*, 2011, **16**: 656-665.

Comment #5

I think it would be important to include the phenotypes of roses overexpressing pre-miR159, which should down-regulate RhCKX6 and increase petal size. This vector is used in the *N. benthamiana* assays.

RESPONSE

Thank you for your excellent advice. We conducted an experiment to investigate the impact of pre-miR159 overexpression on the size of rose petals. The study was based on the methods described in a previous publication (Liang et al., 2020, *Plant Cell*). Our findings revealed that overexpression of pre-miR159 resulted in larger petals compared to control plants and also extended the duration of cell division (from S0 to S2).

We added these results as Supplemental Figure S10.

Supplemental Figure S10 Overexpression of *MIR159* in rose petals.

a,b, Flower opening progression of Control and *MIR159*-OE plants. The experiments were performed independently twice with similar results, and one representative set of results is shown. Scale bars, 2 cm. Data are shown as means \pm SD from 10 biological replicates (n = 10) in **b**.

c, RT-qPCR analysis of miR159 abundance and *RhCKX6* transcript levels in petals of Control and *MIR159*-OE plants. *5S rRNA* was used as an internal control of miR159. *RhUBI2* was used as an internal control of *RhCKX6*. Data are shown as means \pm SD from three biological replicates (n = 3).

d, Petal size of Control and *MIR159*-OE plants at the fully opened stage. Data are shown as means \pm SD from three biological replicates (n = 3). The numbers below the images indicate the petal size. Scale bar, 1 cm. (two-sided Student's *t*-test, *, $P < 0.05$; **, $P < 0.01$).

Comment #6

The authors predict the regulation of several CKXs by miR159 in rose and arabidopsis (Supplementary Fig 4). The location of a miRNA-target site in a target gene is usually conserved during evolution and not variable as depicted in this figure. There is plenty of experimental information identifying miRNA targets in arabidopsis in publicly available databases. Please use this info to show the empirical data for the regulation by miR159 for the two predicted Arabidopsis targets.

RESPONSE

Thank you for your guidance. We searched the public database for the regulation by miR159 for the two predicted Arabidopsis targets. We could find these two targets in the miR159 target survey but we failed to find the change of expression level of these two targets in miR159abc mutant.

To test whether miR159-mediated cleavage of CKXs is a common regulatory pathway, we predicted miR159-target site in *FvCKX6* (*Fragaria vesca*) as well. RLM-RACE confirmed the miR159-mediated cleavage of *FvCKX6* in petals of *Fragaria vesca*, indicating that miR159-mediated cleavage of CKXs might be a common regulation pathway in petals. Supplemental Figure S6 and revised relevant description in the revised manuscript.

Comment #7

Other comments:

Please include in the Supplementary Tables the description for the selected rose genes (or the

description of their arabidopsis homologues).

RESPONSE

Thank you for your advice. We have added the description for the selected genes in *Rosa hybrid*, *Arabidopsis*, *Malus demestica*, and *Fragaria vesca* in Supplementary Table S7.

Comment #8

L78 –“To understand the underlying mechanism of the transition between cell division and cell expansion, we previously conducted a transcriptome deep sequencing (RNA-seq) analysis of short RNAs and transcripts from petals to investigate gene expression profiles during the cell division and expansion phases.” – please include the citation for the article describing the RNAseq analysis of short RNAs and transcripts.

RESPONSE

Thank you for your advice. We have added the citation for describing the RNA-seq analysis of short RNAs and transcripts in the revised Method section.

References

Ben Langmead, Cole Trapnell, Mihai Pop, Steven Salzberg. Ultrafast and memoryefficient alignment of short DNA sequences to the human genome. *Genome Biology*, 2009, **10**: R25.

REVIEWER COMMENTS

Reviewer #1 (Remarks to the Author):

The authors have added the appropriate data and made the appropriate text revisions to address my previous concerns.

Reviewer #2 (Remarks to the Author):

The manuscript by Jing et al. on the role of miR159 in regulating rose petal size has been improved significantly. Most of my points have been addressed. New data have been added which resolves previously open questions. However, I still have some remarks on the previous parts and, in particular, on some of the newly added sections. While the science looks sound to me the description of the results and their interpretation is sometimes not precise enough, which might be due to language problems. I think the authors need help in this matter.

Title

The new title is improved but does in my opinion not match exactly the topic of the manuscript as miR159 appears not to regulate cell division per se but rather the transition from cell division to differentiation. Some suggestions:

In rose, petal size is regulated by the MYB73/TPL/HDA19-miR156-CKX6 module

In rose, petal size is controlled by the MYB73/TPL/HDA19-miR156-CKX6 module regulating (cell number through) cytokinin catabolism

In rose, the MYB73/TPL/HDA19-miR156-CKX6 module regulates the transition from cell division to differentiation and growth and thus petal size

Abstract

Line 22, “earlier cytokinin clearance” rather than “precocious cytokinin clearance”, furthermore an “a” is missing “leading to ‘a’ shortened cell division period”.

Line 23, it is not the cell division itself that is prolonged but the “developmental cell division period”. The reference to cell division instead of cell division period/cell division phase occurs at several occasions in the whole manuscript.

Line 28, Instead of “correct timing of cell division and organ size” it is rather the “correct timing of the exit from the cell division phase and thus the regulation of organ size ...”

Introduction

Line 35, The references do not reflect the mentioned “century-long fascination for size regulation of organisms”.

Line 37, “stepwise“ should be deleted as these phases are dynamically linked

Line 42, the number of cell is determined by the number of divisions, not by the rate of cell proliferation

Line 46, precise what “arrest front” is meant here. Of cell division?

Line 47, cytokinin takes a small “c”

Line 59, “serval” is “several”

There are numerous others of these small mistakes which I do not list all.

Line 83 in a heading it is again not the “duration of cell division” which is controlled by miR159 but the duration of “cell division phase”, and again in line 174, and in line 338

Line 122, I do not see the term “zeatin biosynthesis” in Fig. 2a

Line 207, “cell numbers of cell numbers”, please correct

Line 211, “rapid clearance” was not shown

Discussion

The first part of the Discussion describes size regulation of Arabidopsis leaves and maize by other hormones. While this is interesting it does not fit here, it may be used (in a short version) in the Introduction to describe already known systems of organ size regulation or later in the Discussion as a comparison. Start the Discussion with your own results. It should be concise, highlight the novelties of the findings and put these into perspective.

In the reporting summary under Data analysis: “secondary structure” instead of “second structure”

Reviewer #3 (Remarks to the Author):

The authors conducted additional experiments to generate new data. The revised manuscript has improved significantly.

Specific comments:

Figure S2c: The structure of the precursor is truncated and the miRNA has less than 21nt - more sequence needs to be added.

Fig S8b: The authors show a blot to detect miR319. However, since miR319 and miR159 share a similar sequence, and miR319 is expressed at lower levels compared to miR159, it is likely that the blot is detecting miR159 through cross-hybridization instead of miR319.

Response to Reviewers

Manuscript No. NCOMMS-22-37721

Title: In rose, miR159 regulates petal cell division by modulating cytokinin catabolism

Reviewer #1

COMMENT

The authors have added the appropriate data and made the appropriate text revisions to address my previous concerns.

RESPONSE

Many thanks for your kind comment.

Reviewer #2

COMMENT #1

The manuscript by Jing et al. on the role of miR159 in regulating rose petal size has been improved significantly. Most of my points have been addressed. New data have been added which resolves previously open questions.

However, I still have some remarks on the previous parts and, in particular, on some of the newly added sections. While the science looks sound to me the description of the results and their interpretation is sometimes not precise enough, which might be due to language problems. I think the authors need help in this matter.

RESPONSE

Thank you for your valuable advice. We have carefully revised our manuscript according to your suggestions and asked a professional scientific editing service to enhance the quality of writing and precision of interpretation (<https://planteditors.com/>).

COMMENT #2

The new title is improved but does in my opinion not match exactly the topic of the manuscript as miR159 appears not to regulate cell division per se but rather the transition from cell division to differentiation. Some suggestions:

In rose, petal size is regulated by the MYB73/TPL/HDA19-miR159-CKX6 module.

In rose, petal size is controlled by the MYB73/TPL/HDA19-miR159-CKX6 module regulating cytokinin catabolism.

In rose, the MYB73/TPL/HDA19-miR159-CKX6 module regulates the transition from cell division to differentiation and growth and thus petal size.

RESPONSE

Thank you for your excellent suggestions. We revised the title of our manuscript as 'In rose, petal

size is controlled by the MYB73/TPL/HDA19-miR159-CKX6 module regulating cytokinin catabolism’.

COMMENT #3

Line 22, “earlier cytokinin clearance” rather than “precocious cytokinin clearance”, furthermore an “a” is missing “leading to ‘a’ shortened cell division period”.

RESPONSE

Thank you for your advice. We revised the sentence accordingly. Please see Line 21-22 in the revised version.

COMMENT #4

Line 23, it is not the cell division itself that is prolonged but the “developmental cell division period”. The reference to cell division instead of cell division period/cell division phase occurs at several occasions in the whole manuscript.

RESPONSE

Thank you for your advice. We corrected the relevant description accordingly. Please see Line 23, Line 71, Line 99, Line 189, Line 206, Line 269, Line 314, Line 322, and Line 381 in the revised version.

COMMENT #5

Line 28, Instead of “correct timing of cell division and organ size” it is rather the “correct timing of the exit from the cell division phase and thus the regulation of organ size ...”

RESPONSE

Thank you for your advice. We corrected the description accordingly. Please see Line 28 in the revised version.

COMMENT #6

Line 35, The references do not reflect the mentioned “century-long fascination for size regulation of organisms”.

RESPONSE

Thank you for your advice. We revised this sentence to ‘How the size of living organisms is regulated makes for a fascinating and attractive research question’. Please see Line 33-34 in the revised version.

COMMENT #7

Line 37, “stepwise” should be deleted as these phases are dynamically linked

RESPONSE

Thank you for your suggestion. We have deleted the “stepwise” in Line 36.

COMMENT #8

Line 42, the number of cell is determined by the number of divisions, not by the rate of cell proliferation

RESPONSE

Thank you for your suggestion. We have corrected the description accordingly. Please see Line 40 in the revised version.

COMMENT #9

Line 46, precise what “arrest front” is meant here. Of cell division?

RESPONSE

Thank you for your suggestion. We have corrected the description to 'cell-cycle arrest front' in Line 44 in the revised version.

COMMENT #10

Line 47, cytokinin takes a small "c"

Line 59, "serval" is "several"

RESPONSE

Thank you for your suggestion. We have corrected the description accordingly.

COMMENT #11

Line 83 in a heading it is again not the "duration of cell division" which is controlled by miR159 but the duration of "cell division phase", and again in line 174, and in line 338.

RESPONSE

Thank you for your suggestion. We have corrected the description accordingly. Please see Line 89, Line 99, Line 189, Line 269, Line 314, Line 322, Line 332-333, Line 381, Line 694, Line 752, Line 779-780, Line 824, Line 877, Line 984, Line 1030, and Line 1087 in the revised version.

COMMENT #12

Line 122, I do not see the term "zeatin biosynthesis" in Fig. 2a

RESPONSE

Thank you for your advice. Maybe the font of Fig. 2a is too small. We revised the figure by indicating the term of 'plant hormone signal transduction' and 'zeatin biosynthesis' with an arrow, respectively.

COMMENT #13

Line 207, “cell numbers of cell numbers”, please correct.

RESPONSE

Sorry for this mistake. We corrected it.

COMMENT #14

Line 211, “rapid clearance” was not shown.

RESPONSE

Thank you for pointing this out. We deleted the ‘rapid’. Please see Line 227 in the revised version.

COMMENT #15

Discussion

The first part of the Discussion describes size regulation of Arabidopsis leaves and maize by other hormones. While this is interesting it does not fit here, it may be used (in a short version) in the Introduction to describe already known systems of organ size regulation or later in the Discussion as a comparison. Start the Discussion with your own results. It should be concise, highlight the

novelties of the findings and put these into perspective.

RESPONSE

Thank you for your valuable advice. According to your suggestion, we shortened the discussion of size regulation of Arabidopsis leaves and maize by other hormones, and moved this part to Introduction section in the revised version.

COMMENT #16

In the reporting summary under Data analysis: “secondary structure” instead of “second structure”

RESPONSE

Thank you for pointing this out. We have corrected the description in Date analysis of the reporting summary.

Reviewer #3

COMMENT #1

Figure S2c: The structure of the precursor is truncated and the miRNA has less than 21nt-more sequence needs to be added.

RESPONSE

Thank you for your valuable advice. We re-predicted the structure of *rhy-MIR159* containing the 21nt of miR159 in the revised Figure S2c.

c

COMMENT #2

Fig S8b: The authors show a blot to detect miR319. However, since miR319 and miR159 share a similar sequence, and miR319 is expressed at lower levels compared to miR159, it is likely that the blot is detecting miR159 through cross-hybridization instead of miR319.

RESPONSE

Thank you for pointing this out.

To specifically detect miR159 and miR319, we used Locked Nucleic Acids (LNA)-modified oligonucleotide probes to ensure the specificity of Northern blot of miR159 and miR319 (Válóczi et al., 2004). LNA is a class of bicyclic high-affinity RNA analogues in which the furanose ring in the sugar-phosphate backbone is chemically locked in an N-type (C3'-endo) conformation by the introduction of a 2'-O,4'-C methylene bridge (Obika et al., 1997; Koshkin et al., 1998). The unprecedented thermal stability of LNA oligonucleotides together with their improved mismatch discrimination ensure the highly specificity of LNA-modified probes.

miR159 probe: 5'-taGAGCTCCCTTCAATCCAAa-3'

miR319 probe: 5'- ggAGCTCCCTTcAGTCCAAt-3'

The lowercase letters indicate the LNA-modified oligonucleotide.

In our previous results of Northern blot of miR319, we prolonged the exposure time to obtain the optimum image. To compare the expression level of miR159 and miR319, we re-conducted the Northern blot by using same exposure time (20 min) for both miR159 and miR319 in petals on 1, 3, and 5 days after stage 0. As shown in the below figure, the expression level of miR319 was much lower than miR159. We used these results as Fig S8b, and added relevant description of LNA probes in Legend of Figure 1a and Fig S8b in the revised version.

Reference:

- Válóczi A., Hornyik C., Varga N., Burgyán J., Kauppinen S., Havelda Z., 2004. Sensitive and specific detection of microRNAs by northern blot analysis using LNA-modified oligonucleotide probes. *Nucleic Acids Res*, 32: 175-175.
- Obika S., Nanbu D., Hari Y., Morio K., In Y., Ishii J.K., Imanishi T., 1997. Synthesis of 2'-O,4'-C methylneuridine and cytidine. Novel bicyclic nucleosides having a fixed C3-endo sugar puckering. *Tetrahedron Lett*, 38: 8735-8738.
- Koshkin A.A., Singh S.K., Nielsen P., Rajwanshi V.K., Kumar R., Meldgaard M., Olsen C.E., Wengel J., 1998. LNA (locked nucleic acids): synthesis of the adenine, cytosine, guanine, 5-methylcytosine, thymine and uracil bicyclonucleoside monomers, oligomerisation, and unprecedented nucleic acid recognition. *Tetrahedron*, 54: 3607-3630.